# Single-molecule force spectroscopy reveals the dynamic strength of the hair-cell tip-link connection

Eric M. Mulhall[1,2], Andrew Ward [1,3], Darren Yang [3], Mounir A. Koussa[1,2], David P. Corey [1,6✉] & Wesley P. Wong [3,4,5,6✉]

The conversion of auditory and vestibular stimuli into electrical signals is initiated by force transmitted to a mechanotransduction channel through the tip link, a double stranded protein filament held together by two adhesion bonds in the middle. Although thought to form a relatively static structure, the dynamics of the tip-link connection has not been measured. Here, we biophysically characterize the strength of the tip-link connection at single-molecule resolution. We show that a single tip-link bond is more mechanically stable relative to classic cadherins, and our data indicate that the double stranded tip-link connection is stabilized by single strand rebinding facilitated by strong cis-dimerization domains. The measured lifetime of seconds suggests the tip-link is far more dynamic than previously thought. We also show how $Ca^{2+}$ alters tip-link lifetime through elastic modulation and reveal the mechanical phenotype of a hereditary deafness mutation. Together, these data show how the tip link is likely to function during mechanical stimuli.

[1] Department of Neurobiology, Blavatnik Institute, Harvard Medical School, Boston, MA, USA. [2] Program in Neuroscience, Harvard University, Cambridge, MA, USA. [3] Program in Cellular and Molecular Medicine, Boston Children's Hospital, Boston, MA, USA. [4] Department of Biological Chemistry and Molecular Pharmacology, Blavatnik Institute, Harvard Medical School, Boston, MA, USA. [5] Wyss Institute for Biologically Inspired Engineering, Harvard University, Boston, MA, USA. [6] These authors jointly supervised this work: David P. Corey, Wesley P. Wong. ✉email: dcorey@hms.harvard.edu; wesley.wong@childrens.harvard.edu

Our senses of hearing and balance rely on the extraordinarily sensitive molecular machinery of the inner ear to convert deflections as small as the width of a single carbon atom into electrical signals that the brain can process[1–3]. In humans and other vertebrates, these senses are mediated by hair cells, named for the bundle of hair-like stereocilia protruding from their apical surfaces (Fig. 1a). Deflection of this organelle during a mechanical stimulus tenses tip links, the thin, 170-nm-long protein filaments that connect the tip of each stereocilium to the lateral wall of its tallest neighboring stereocilium[4,5]. Tip links bind to and directly gate mechanotransduction channels at their lower ends (Fig. 1a)[6–9].

The tip link is a double-stranded complex of the $Ca^{2+}$-dependent extracellular adhesion proteins protocadherin-15 (PCDH15) and cadherin-23 (CDH23). A parallel dimer of CDH23 binds to a parallel dimer of PCDH15 at their N termini to form an antiparallel helical tetramer (Fig. 1a)[10,11]. CDH23 is a long atypical cadherin comprising the upper 2/3 of the tip link; it has 27 extracellular (EC) domains, a specialized membrane-adjacent domain, a single-pass transmembrane helix, and an intracellular domain[12]. The CDH23 intracellular domain binds to the scaffolding proteins harmonin (USH1C) and sans (USH1G), which in turn bind to myosin motors that move along the actin core filaments of stereocilia[13,14]. These myosin motors are likely responsible for the millisecond-timescale adaptation of transduction current and for maintaining the resting open probability of the transduction channel[15–17]. PCDH15 has a similar domain structure to CDH23 but contains only 11 EC domains[18]. PCDH15 binds to several components of the mechanotransduction apparatus at the lower end of the tip link, including the pore-forming subunits of the mechanotransduction channel TMC1 and TMC2[6]. Exactly how tensile force from PCDH15 is transmitted to the mechanotransduction channel is unknown, but the N-terminal binding domains of PCDH15 and CDH23 must be connected in *trans* in order to convey mechanical information[7,10,19].

While classic *trans* cadherin bonds are mediated by homophilic strand swapping of an N-terminal tryptophan residue into a

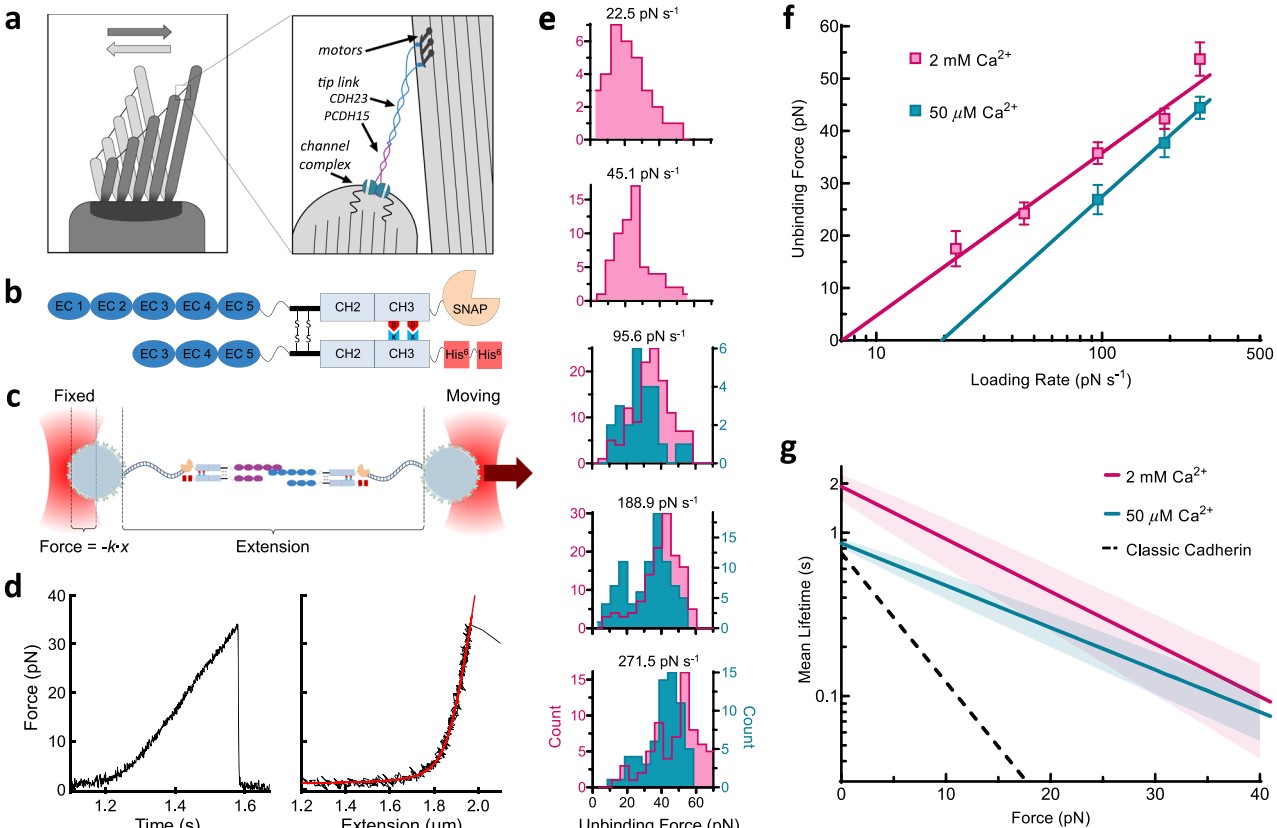

**Fig. 1 The mechanical strength of a single tip-link bond. a** The hair-cell stereocilia bundle and mechanotransduction complex. Bundle deflection to the right increases tension on tip links, composed of CDH23 and PCDH15. Here and throughout, PCDH15 is shown in purple and CDH23 in blue. **b** Single-bond fusion proteins containing one EC1-2 binding domain. Antibody Fc domains (CH2, CH3) fused to the EC domains bind to create a dimer, masking non-specific binding of EC3-5. The SNAPtag binds to the DNA tether. **c** Single-bond fusion proteins in the dual-beam optical trap. Force was calculated as a linear function of displacement from a stiffness-calibrated optical trap. The extension was measured as the distance between the surfaces of the two microspheres. **d** Representative force-time and force-extension profiles for a single-bond unbinding event. The force-time profile was used to extract the force-loading rate in pN s$^{-1}$ in the linear regime preceding bond rupture. The force-extension profile was fit with an extensible worm-like chain (WLC) model[95] to extract contour length, persistence length (related to bending stiffness), and unbinding force for each unbinding trace. **e** Histograms of unbinding forces at different force-loading rates in 50 μM (cyan) and 2 mM (pink) $Ca^{2+}$. **f** Most-probable unbinding forces plotted as a function of loading rate. Systematic kernel density estimation was used to determine the most likely unbinding force for each condition. Error is shown as the optimal bandwidth from a kernel density estimation. A weighted linear fit of the data with the Bell–Evans model was used to extract force-dependent unbinding kinetics[39], where $f_\beta$ is the slope and $k_{off}^0 \cdot f_\beta$ is the x-intercept. For 50 μM $Ca^{2+}$, $n = 174$ unbinding events, and for 2 mM $Ca^{2+}$ $n = 411$ unbinding events. **g** Mean single-bond lifetime as a function of force, calculated using the intrinsic zero-force off rate $k_{off}^0$ and the force scale $f_\beta$ from **f**, and compared to the average lifetime of classic cadherin bonds (dashed black line)[21–25]. The propagated errors in lifetime due to errors in fit parameters from **f** are shown as light bands.

hydrophobic pocket of its binding partner[20], we previously demonstrated that the heterophilic *trans* tip-link bond has a far more extensive, amphiphilic "handshake" interface involving ~30 amino acids in the first two EC domains of each protein[11]. Yet, despite having a far more extensive interface, the equilibrium dissociation constant of a single tip-link bond is nearly the same as for classic cadherin *trans* bonds[11,21–25]. Mutations that occur in the bond interface of each protein are known to alter the equilibrium dissociation constant ($K_D$) of the interaction[11,26] and to cause deafness in humans and other animals[27–29]. However, dissociation constants provide little insight into how the tip link functions in response to dynamic force stimuli and how these mutations result in deafness. Tip links experience a ~10-piconewton (pN) resting tension exerted by myosin motors[12,17,30], they may experience forces up to 100 pN[31], and the force oscillates at over 100 kHz in some species[32]. Therefore, examining the behavior of the tip link under tension is critical for describing its function, especially in the context of disease-causing mutations.

The tip link's double-stranded quaternary structure is also distinct from classic cadherins, which form clusters of tens or hundreds of monomeric proteins in a variety of tissues[33] and which rely on the additive strength of many links to maintain their connection over hours, days, or weeks[34,35]. The tip link, with just two strands, must be strong enough to continuously convey mechanical information to the mechanotransduction channel under both the resting tension and the dynamic forces exerted by sound or head movement. However, since tip links are vulnerable to acoustic trauma in vivo[36,37], they apparently release under a high-intensity stimulus. How does the tip link perform these dual functions?

Here, we employed single-molecule force spectroscopy, biochemistry, and kinetic modeling to quantitatively characterize the dynamics of the tip-link connection. We show the tip-link connection has multiple mechanisms—distinct from classic cadherin linkages—that facilitate mechanotransduction in the inner ear. These mechanisms include an increased resistance to mechanical force relative to classic cadherins, a double-stranded architecture mediated by strong *cis*-dimerization that permits rebinding before rupture, and $Ca^{2+}$-dependent elastic properties that modify the lifetime of the connection. We also show how a hereditary deafness mutation gives rise to a unique mechanical phenotype through modulation of the bond's sensitivity to force. From these measurements, we predict that the tip-link lifetime, in response to oscillatory stimuli, is largely insensitive to a wide range of hair-cell deflections within the physiological range. We estimate the lifetime to be on the order of seconds, far shorter than previously predicted, suggesting that there is rapid formation of new tip-link connections. These results provide a depiction of the strength and the dynamics of the tip-link connection and provide a foundation for further studies of its role in transducing mechanical information in the inner ear.

## Results

**The mechanical strength of a single tip-link bond**. To measure the mechanical strength of tip-link cadherins, we used optical tweezers to perform single-molecule dynamic force spectroscopy on engineered fusion proteins (Fig. 1b–d and Supplementary Fig. 1). Fusion proteins were expressed in and purified from mammalian cells; they included between 3 and 27 EC domains of PCDH15 or CDH23, fused to an antibody Fc-domain to promote dimerization. Paired Fc domains were then linked to one end of a DNA tether through a SNAPtag. DNA tethers were biotinylated at the other end and linked to streptavidin-coated silica microspheres of ~3-μm diameter, passivated with a polyethylene glycol (PEG) brush (Supplementary Fig. 1a–e). These beads, attached to

either CDH23- or PCDH15-linked DNA–protein complexes and distinguishable via a fluorescent marker, were placed in a flow cell on the microscope stage, and manipulated with a dual-beam optical tweezers setup (see "Methods" section). They were brought close together, to allow a PCDH15–CDH23 bond to form, and then moved apart while recording extension and force. As the force built up the bond eventually ruptured, a transition that was easily identified by the abrupt drop to zero-force (Fig. 1d and Supplementary Fig. 1f). The use of long DNA tethers reduced non-specific interactions and enabled the identification of each tether as the desired single-molecule linkage based on force-extension curves (Supplementary Fig. 1g, h).

We first studied the unbinding of single bonds, by engineering asymmetric dimers in which only one strand contained the EC1-2 binding domain (Fig. 1b, c). We found that when beads were moved apart more rapidly (i.e., a faster force-loading rate was applied to the bond), the bond broke at higher forces (Fig. 1e)—as expected if it takes some time to unbind and the force is increasing during that time. The most-probable unbinding force increased linearly with the logarithm of the loading rate, indicating that force also accelerated the unbinding rate (Fig. 1f). Force often destabilizes adhesion bonds by lowering the energy barrier for unbinding, decreasing the time that two proteins will remain bound[38]. To characterize key variables for the force-dependent unbinding, we fitted unbinding force data with the Bell–Evans model (Fig. 1e, f)[39]. First, we found that the mean lifetime of the single tip-link bond at zero-force ($1/k_{off}^0$) in 2 mM $Ca^{2+}$ is $1.9 \pm 0.4$ s ($k_{off}^0 = 0.5 \pm 0.1$ s$^{-1}$) (Fig. 1g), agreeing well with independent zero-force measurements[11,26,39]. Second, we found that the force sensitivity of the tip-link bond is $f_\beta = 13.5 \pm 2.0$ pN (Fig. 1f). This fit parameter of the Bell–Evans model is the characteristic force required to accelerate the rate of dissociation $k_{off}$ by $e$-fold. Both the linear relationship between the most-probable rupture forces and the logarithm of the force-loading rate, and the consistency between the independently measured zero-force off rate and that calculated from our data, indicate that the tip-link bond has traditional slip-bond behavior[39]. Compared with classic *trans* cadherin bonds (mean zero-force lifetime ≈ 0.75 s, $f_\beta \approx 5.5$ pN)[21–25], the single tip-link bond has a 2.5-fold longer lifetime at zero-force, and the unbinding rate is significantly less sensitive to mechanical force (Fig. 1g).

The tip-link bond is destabilized by very low $[Ca^{2+}]$[11,19], so we measured unbinding in a $Ca^{2+}$ concentration near that of the endolymph fluid that bathes tip links in the cochlea. However, we found that the mean lifetime of a single tip-link bond at zero-force is reduced by only 50% in 50 μM $Ca^{2+}$ (Fig. 1g). This insensitivity to low $[Ca^{2+}]$ is in stark contrast with classic cadherins, for which half-maximal adhesion of even the relatively $Ca^{2+}$-insensitive N-cadherin is observed at 720 μM $Ca^{2+}$[40]. Thus, the tip-link bond is less sensitive to the $Ca^{2+}$-poor cochlear and vestibular endolymph, which enables the tip link to reliably convey mechanical stimuli in the inner ear.

**The dynamics of the double-stranded tip-link connection**. Why is the tip link double-stranded? Avidity is the cumulative strength that results from multiple bonds, and one important contribution to avidity is the division of externally applied force among the individual components[41]. Force-sharing among bonds can result in dramatic increases in lifetime because the off-rate of a single bond has an exponential dependence on force. Another contribution comes from the ability of individual components to rebind after they rupture, dynamically maintaining the overall connection. Since CDH23 and PCDH15 both form strong *cis*-dimers in vitro and in vivo, facilitated by bonds along their ectodomains[5,10,42–45], we hypothesized that the tip link could

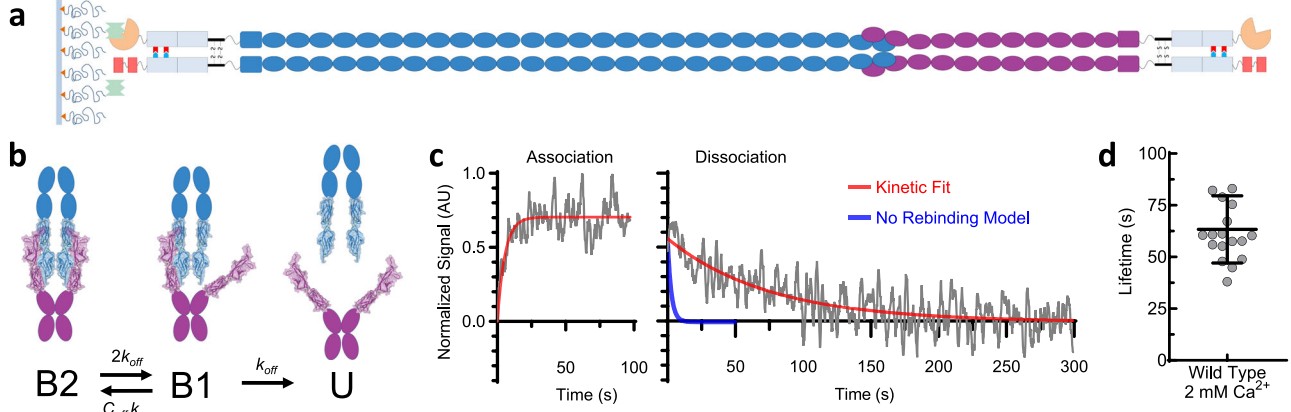

**Fig. 2 The dynamics of the double-stranded tip-link connection. a** Schematic of dimeric, full-ectodomain proteins, which were used for both biolayer interferometry and force-spectroscopy experiments, attached to the biolayer-interferometer sensor surface. Fc domains, SNAPtag, and tandem His6 tag as in Fig. 1b. **b** State diagram for tip-link avidity during dissociation. B2 is the doubly bound state, B1 the singly-bound, and U is unbound. $k_{off}$ is the single-bond off-rate. The rebinding rate B1→B2 depends on the intrinsic on-rate $k_{on}$ and the effective concentration $C_{eff}$, which is determined by the volume through which the unbound ends can move and remain in proximity. The N-terminal protein structure shown in the diagram is adapted from the Protein Data Bank (PDB) structure 4AXW. **c** Sample biolayer interferometry traces for dimeric full-ectodomain PCDH15 and CDH23 fusion proteins. An incident white light transmitted through the fiber-optic sensor was differentially reflected from the glass sensor surface and from the bound protein layer (here CDH23); changes in the protein layer upon binding of a soluble protein (here PCDH15) produced the binding signal. (Left) Association timecourse enables calculation of $C_{eff} k_{on}$. (Right) The dissociation timecourse of the dimer (fit in red) indicates a long lifetime relative to the expected lifetime in the absence of rebinding (calculated in blue). Here, 320 nM of both PCDH15 and EC1-2 truncated PCDH15 were used for the association phase (Supplementary Fig. 2c). **d** Scatter plot of tip-link connection lifetimes measured with biolayer interferometry at 2 mM Ca$^{2+}$ (63 ± 16 s, mean ± SD, $n = 18$ independent experiments).

dramatically increase its lifetime through rebinding of unbound PCDH15 and CDH23 EC1-2 domains.

The tip-link connection may be described by a kinetic model (Fig. 2b and see "Methods" section) in which a transition from the double-bound (B2) to unbound (U) state requires an intermediate single-bound (B1) state. If opposing PCDH15 and CDH23 EC1-2 domains are spatially constrained by *cis*-dimerization interfaces[42], the effective concentration might be sufficiently high and the concentration-dependent binding rate $C_{eff} k_{on}$ of an unbound pair might then be sufficiently fast, relative to the single bond off-rate $k_{off}$, to allow the bond to transition back to the B2 state rather than moving to the U state where complete rupture occurs. One approximation of the effective local concentration $C_{eff}$ can be obtained by calculating the volume of a sphere with a radius equal to the length of an individual EC1-2 binding domain (~10 nm)[11], yielding $C_{eff}$ ~400 μM (see "Methods" section). Although N terminal *cis*-dimerization interfaces for CDH23 are not yet solved, a *cis*-dimerization interface is present at PCDH15 EC3[42], consistent with our estimate of sphere size. Using a single-bond $k_{on}$ of 6.2 ± 2.8 ×10$^4$ M$^{-1}$ s$^{-1}$ [46] and a zero-force single-bond $k_{off}^0$ of 0.5 s$^{-1}$ (Fig. 1g), we estimated the zero-force rebinding rate (B1 → B2) to be ~25 ± 14 s$^{-1}$ and the overall zero-force lifetime of the connection to be 22–63 s (see "Methods" section), much longer than the single-bond zero-force lifetime, which we determined to be 1.9 ± 0.4 s.

To test the hypothesis that rebinding results in a substantially increased lifetime at zero-force, we used biolayer interferometry[47,48] and fusion proteins containing the entire dimerized extracellular domains of PCDH15 and CDH23 to measure the zero-force kinetics of the double-bond tip link connection (Fig. 2a, c, d and Supplementary Fig. 2). We found the effective on-rate to be 3.5 ± 0.7 × 10$^5$ M$^{-1}$ s$^{-1}$, in reasonable agreement with published single-bond on-rates[46], and the mean lifetime of the connection to be 63.3 ± 3.8 s (SEM, $n = 18$), in good agreement with the lifetime estimated from force spectroscopy. These values provide an empirical means of determining the effective concentration $C_{eff}$ in the B1 state, yielding $C_{eff} = 650 ± 200$ μM. This is three orders of magnitude more concentrated than the equilibrium $K_D$ of a single-

bond interaction[11,26], consistent with our estimates and sufficient for promoting rapid rebinding to the B2 state. Thus, our data suggest that at zero-force the lifetime of the tip-link connection is substantially increased by rebinding, and rebinding is likely facilitated by *cis*-dimerization domains which keep binding domains in close proximity and the effective concentration high.

**The strength of the tip-link connection under force.** To determine the dynamics of the tip-link connection under force, we used optical tweezers to measure the unbinding kinetics of the double-stranded tip link containing the entire cadherin ectodomains (Figs. 2a and 3a–c and Supplementary Table 1) (see "Methods" section). We found a large increase in strength (Fig. 3a, black circles) relative to the single bond (magenta squares). From single-bond data (Fig. 1f), we can independently predict the rupture force in the case of simple load sharing between two bonds[41] (Fig. 3a, "No Rebinding", and "Methods" section). This line represents dimer strength with perfect load sharing between the two strands, and with a unidirectional unbinding transition through the states B2 → B1 → U; it is the maximum strength achievable from two independent slip bonds without rebinding.

Within the loading-rate range of 0.39 to 22.5 pN s$^{-1}$, the measured unbinding forces were significantly higher than predicted by perfect load sharing (Fig. 3a). We wondered whether atypical single-bond properties might explain this increase in most likely rupture force. Although we were unable to make reliable single-bond optical tweezer measurements at loading rates below 22.7 pN s$^{-1}$, an alternative method to calculate bond lifetimes at constant forces fit well with our single-bond model and showed no deviation from slip bond behavior, even at very low forces (Supplementary Fig. 3a)[49,50]. Importantly, each method predicts a bond lifetime at zero-force of ~2 s, consistent with independent zero-force measurements[26]. Another possible explanation for an increased dimeric lifetime is bond cooperativity; however, full cooperativity predicts an unreasonably low energy barrier for a single-bond and a loading-rate dependence

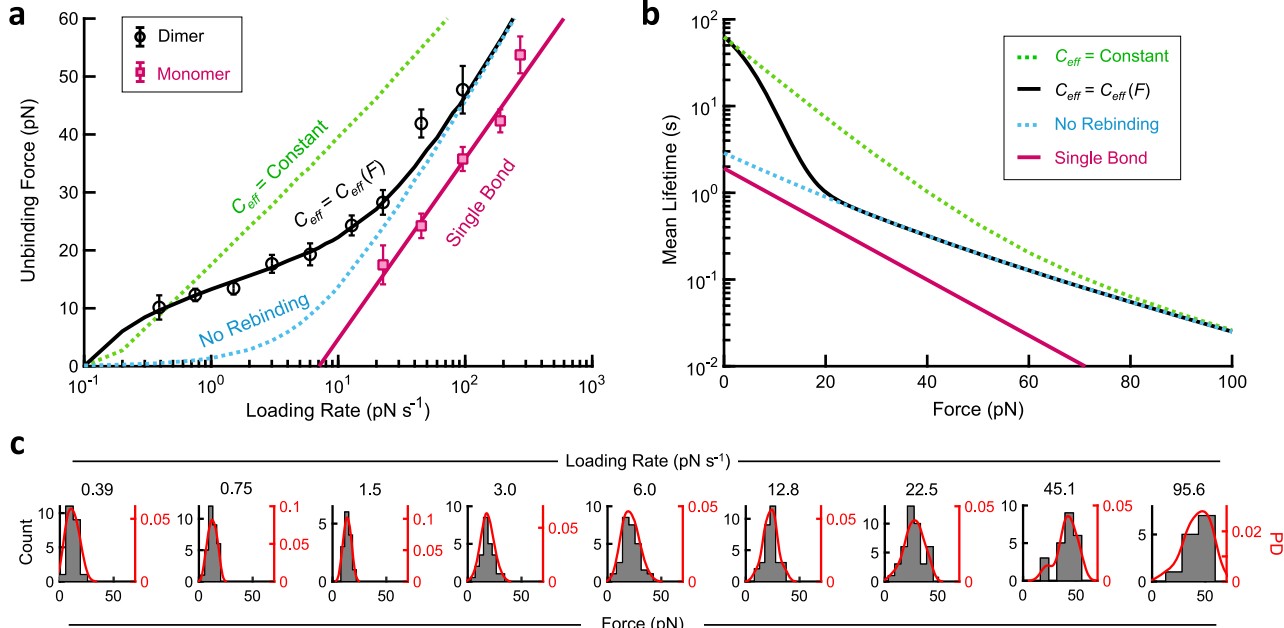

**Fig. 3 The strength of the tip-link connection under force. a** Unbinding force as a function of loading rate for full-length dimers in 2 mM Ca$^{2+}$. Displayed are most-probable unbinding forces for double-bond full-length dimer proteins (black circles, $n = 247$ unbinding events) and single-bond unbinding forces from Fig. 1f (magenta squares, $n = 411$ unbinding events). Error is shown as the optimal bandwidth from a kernel density estimation. Dimer unbinding forces show enhanced strength relative to single bonds, particularly at slow loading rates. A kinetic model that incorporates rebinding was fitted to the double-bond interaction (solid black line). A no-rebinding model is shown as the blue dashed line; a rebinding model with constant $C_{eff}$ is shown as the green dashed line. **b** Calculated mean lifetime from the model fits in **a**. At forces below about 20 pN, the tip link connection has a pronounced increase in lifetime relative to the single bond due to avidity. At low forces, this results from a combination of force-sharing and rebinding. At high force, unbinding becomes faster and rebinding becomes slower, so load sharing dominates, resulting in half the sensitivity to force relative to a single bond. **c** Histograms of unbinding forces plotted in **a**. Unbinding forces at each loading rate were systematically binned using the Freedman–Diaconis rule to yield $N$ total bins of width $\Delta F$. Bin centers were systematically chosen to yield the most likely unbinding force with the maximum number of counts. A kernel smoothing function density estimate is overlaid on the binned unbinding force data.

for bond rupture that is incompatible with our force spectroscopy data[41].

A rebinding model, based on single-bond kinetics and a constant $C_{eff}$ that was determined from biolayer interferometry, successfully predicts higher unbinding forces relative to no rebinding, but it predicts forces even higher than observed (Fig. 3a, "$C_{eff}$ = Constant"). What can account for this? Recent studies using optical tweezers and molecular dynamics simulations suggest that tip link proteins are far more elastic than initially thought[51,52]. Elasticity would allow unbound domains in the B1 state (Fig. 2a) to separate as force increases, decreasing $C_{eff}$ and reducing rebinding. A three-state kinetic model, in which $C_{eff}$ decreases with a Gaussian decay dependent on the compliance parameter $f_c$ (see "Methods" section)[50], fits the data well (Fig. 3a, "$C_{eff}$ = $C_{eff}$ (F)"), yielding $f_c = 10.0 \pm 1.8$ pN and an effective concentration at zero-force of $C_{eff}^0 = 465 \pm 100$ μM (Supplementary Fig. 3c). This model predicts a zero-force mean lifetime of $62 \pm 17$ s, in excellent agreement with both our biolayer interferometry measurements and our estimates based on extrapolating the effect of avidity from single-bond measurements. The model also indicates that protein elasticity can be an important factor in determining the dynamics of the tip-link connection within the physiological range of forces.

Importantly, these measurements enable the prediction of mean tip-link connection lifetimes as a function of force (Fig. 3b and Supplementary Fig. 3a) for different force histories. At a constant resting tension of 10 pN[53], the double-bond interaction lasts ~9 s, an order of magnitude longer than a single-bond interaction of 0.9 s. Yet at very high forces of 50–60 pN the predicted lifetime is much less than a second. The tip link has

apparently evolved to use avidity to stabilize the connection at the normally low operating forces but releases exponentially faster at higher forces.

**Cis-dimerization stabilizes avidity of the tip-link connection under force.** Our model predicts that the mechanical and structural properties of tip-link cadherins significantly affect rebinding and overall avidity. One contributor to these mechanical properties is the *cis*-dimerization that links dimers of PCDH15 and of CDH23 laterally at several points along their extracellular domains[5,10,43–45]; indeed, disruption of a *cis*-dimerization domain at EC3 of PCDH15 disrupts mechanotransduction[42]. For a full-length tip-link protein, unbinding of one PCDH15–CDH23 bond in the tip-link connection would redistribute tension to the strand with the remaining bond (Fig. 2b, Supplementary Fig. 3c). If the two strands were essentially independent, the loaded strand would stretch more than the unloaded strand, causing shearing between the strands. In the B1 state, strand shearing is likely to result in an increased distance between the unbound EC1–EC2 domains of PCDH15 and CDH23, reducing $C_{eff}$, decreasing the rebinding rate $C_{eff}k_{on}$ under force loads, and consequently decreasing the lifetime of the connection overall. However, *cis*-dimerization at multiple points could reduce shearing and facilitate rebinding to prolong bond lifetime. These shearing and force distribution phenomena are well described for DNA, actin, and other biopolymer filaments containing multiple dimerization domains[54–57].

To disrupt full *cis*-dimerization and explore this effect, we truncated PCDH15 and CDH23 to their first five N-terminal EC domains and used single-molecule force spectroscopy to

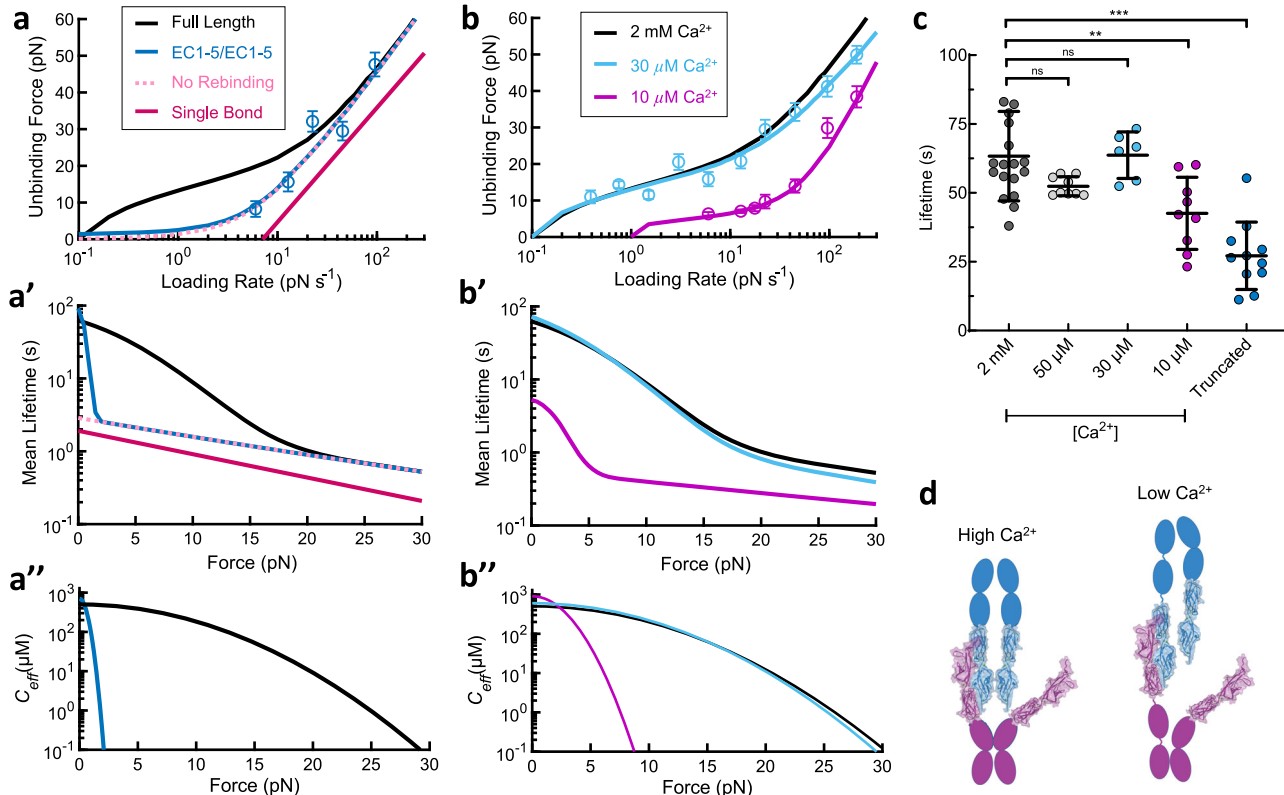

**Fig. 4 _Cis_-dimerization and extracellular $Ca^{2+}$ alter the lifetime of the tip-link connection through elastic modulation. a** Avidity under force is enhanced by _cis_-dimerization. Dimeric tip-link proteins were truncated to just the first five EC domains, to remove _cis_-dimerization interfaces further from the N termini. Unbinding forces (blue circles, $n = 159$ unbinding events) were fit with the force-dependent avidity model, with fixed parameters $k_{off} = 0.5\,s^{-1}$ and $f_\beta = 13.5\,pN$ (EC1-5/1-5). Error is shown as the optimal bandwidth from a kernel density estimation. At the rupture forces tested, the model fit was indistinguishable from a model in which there is no rebinding. **a′** Truncated proteins show an enhancement in lifetime above 6 pN primarily attributable to force-sharing. **a″** $C_{eff}$ calculated from the fit decreased rapidly with the applied force for EC1-5 dimers. **b** Low $[Ca^{2+}]$ destabilizes the tip link. Unbinding forces in 30 μM $Ca^{2+}$ (blue, $n = 339$ unbinding events) were nearly identical to those in 2 mM $Ca^{2+}$ (black), but were much lower in 10 μM $Ca^{2+}$ (magenta, $n = 344$ unbinding events). Error is shown as the optimal bandwidth from a kernel density estimation. Model fits suggested that 10 μM $Ca^{2+}$ weakened the connection both by accelerating the single-bond off-rate ($k_{off} = 2.6 \pm 0.8\,s^{-1}$) and by altering the mechanical properties of the protein complex ($f_c = 2.9 \pm 1.1\,pN$). **b′** Calculated lifetimes of the tip link in 30 μM $Ca^{2+}$ at forces between 0 and 30 pN were indistinguishable from lifetimes at 2 mM $Ca^{2+}$ but were severely decreased at 10 μM $Ca^{2+}$. **b″** Calculated $C_{eff}$ decreased rapidly with applied force at 10 μM $Ca^{2+}$ through increased protein elasticity. **c** Scatter plot of full-length dimer lifetimes measured from biolayer interferometry experiments at various concentrations of $Ca^{2+}$, and of the truncated EC1-5/EC1-5 dimer in 2 mM $Ca^{2+}$ (mean ± SD). 2 mM: 63 ± 16 s ($n = 18$ independent experiments), 50 μM: 52 ± 4 s ($n = 9$ independent experiments), 30 μM: 64 ± 9 s ($n = 6$ independent experiments), 10 μM: 43 ± 13 s ($n = 9$ independent experiments), Truncated: 27 ± 12 s. The Student's two-tailed unpaired _t_-test was used to determine statistical significance (**$p = 0.0028$, ***$p < 0.001$). **d** Schematic of singly-bound dimeric tip links, illustrating how an increase in elasticity produced by low extracellular $Ca^{2+}$ can separate unbound ends and lower $C_{eff}$.

determine their strength (Fig. 4a). At higher loading rates where rebinding is not expected to be significant (>20 pN s⁻¹), truncated EC1-5 dimers still displayed an enhanced binding strength relative to the single-bond interaction, very similar to the full dimer proteins. This indicates that truncation did not disrupt the attachment of both EC1-2 binding domains. However, at slower loading rates where rebinding is expected to significantly impact kinetics, the truncated EC1-5 dimer ruptured at lower forces than the full dimer, exhibiting a lifetime similar to that predicted by a no-rebinding model with perfect force sharing (Fig. 4a, pink dashed line). A fit of our force-dependent concentration rebinding model to the EC1-5 data (Fig. 4a, blue line) was not different from the no-rebinding model and the predicted zero-force lifetime had a high error (Supplementary Table 1). Therefore, we conclude that although the truncated EC1-5 dimer binds with both EC1-2 binding domains, rebinding does not appear to occur at forces above 6 pN over the range of loading rates we tested. But what happens under conditions of low-to-zero stress?

To measure the lifetime of the EC1-5 tip-link connection at zero-force and the potential influence of rebinding under these conditions, we used biolayer interferometry as before. We found the lifetime to be 27.1 ± 3.7 s (SEM, $n = 11$) (Fig. 4c), approximately half as long as the full-length tip-link connection (63.3 ± 3.8 s, Fig. 2d), but still over an order of magnitude longer than a single bond (1.9 ± 0.4 s, Fig. 1g), indicating rebinding is still occurring but to a lesser extent. We calculated that unbound strands of EC1-5 tip-link connection have an effective concentration for rebinding $C_{eff}$ of ~170 μM at zero-force (see "Methods" section), compared with ~650 μM for the full-length tip-link connection. From the force spectroscopy data, we calculate that at forces as low as 6 pN, $C_{eff}$ drops to essentially zero for the EC1-5 dimer but is maintained at ~350 μM for the full-length tip-link connection (Supplementary Fig. 3b). This is consistent with the idea that the loss of _cis_-dimerization domains in the truncated EC1-5 protein allows for more shearing under force and represents a likely explanation for the observed increase in force sensitivity $f_c$ of the EC1-5 dimer. These data highlight the

importance of full *cis*-dimerization in enhancing the strength and lifetime of the tip-link connection relative to a single tip-link bond.

**Extracellular Ca²⁺ alters tip-link lifetime through elastic modulation**. $Ca^{2+}$ ions link the EC domains in each tip-link protein[11,58], and act to maintain its structural integrity[51]. Chelation of extracellular $Ca^{2+}$ disrupts tip links in intact hair cells and destabilizes the CDH23–PCDH15 interaction (Figs. 1g and 4c)[11,19]. Molecular dynamics simulations suggest that as the concentration of $Ca^{2+}$ bathing the tip link decreases and there is decreased occupancy of $Ca^{2+}$-binding sites, the junctions between each EC-domain become more flexible and the series elasticity of EC domains is increased[52,58]. Because protein elasticity is important in determining the extent of avidity under force (Figs. 3a and 4a), increased compliance in low $Ca^{2+}$ should cause unbound EC domains in the B1 state (Fig. 2b) to separate more rapidly as force increases, decreasing $C_{eff}$ and reducing rebinding.

We, therefore, measured unbinding forces in low $[Ca^{2+}]$ (Fig. 4b). In 30 μM $Ca^{2+}$, the concentration found in bulk cochlear endolymph[59,60], the lifetime of the tip-link connection at all forces was not significantly altered relative to 2 mM $Ca^{2+}$, again demonstrating that the tip link is well equipped to convey mechanical stimuli in a physiological environment. Consistent with these results, tip-link lifetime at zero-force was not significantly decreased in 50 or 30 μM $Ca^{2+}$ when measured with biolayer interferometry (Fig. 4c). However, at the sub-endolymphatic level of 10 μM $Ca^{2+}$, the tip-link connection was severely weakened under force when measured with force spectroscopy (Fig. 4b) and significantly destabilized when measured with biolayer interferometry (Fig. 4c). Based on model fits to our force spectroscopy data, the single-bond off-rate at this calcium concentration is accelerated ~4 fold and the protein stiffness is decreased ~10 fold (Supplementary Table 1), causing $C_{eff}$ to decrease rapidly with force. Thus, depletion of extracellular $Ca^{2+}$ modifies the lifetime of the tip-link connection through two distinct mechanisms: the intrinsic single-bond off rate, and the stiffness-dependent decrease in $C_{eff}$.

**The mechanical phenotype of a hereditary deafness mutation**. Hearing loss is one of the most common congenital disorders, and more than half of all cases are due to genetic causes[61]. A broad spectrum of Usher syndrome mutations disrupts tip-link integrity in hair cells[62,63], including the human R113G mutation in *PCDH15*, which causes hearing loss without a vestibular or visual phenotype[28]. The R113 residue participates in the binding interface (Fig. 5b), and the R113G mutation reduces bond affinity in vitro[11,46]. To determine how this mutation disrupts function, we used polarized Fc-domain dimerization (see "Methods" section) to introduce this mutation either heterozygously (R113G⁺/⁻) or homozygously (R113G⁺/⁺) into full-ectodomain mouse PCDH15 proteins and measured their strength with force spectroscopy. We found that the heterozygous R113G⁺/⁻ tip-link bond (brown circles) is weaker than the wild-type, and the homozygous R113G⁺/⁺ bond (purple circles) is much weaker at all forces (Fig. 5c, d). The exceptional reduction in strength by a single amino acid substitution is additional confirmation that our measurements are specific to the tip-link bond.

To understand how this mutation disrupts function, we performed a simultaneous fit of R113G⁺/⁻ and R113G⁺/⁺ unbinding forces, using modified three- and four-state force-dependent models (Fig. 5a and see "Methods" section). Most strikingly, the R113G mutation causes the single-bond off-rate to be ~8 times more sensitive to force ($f_\beta^{R113G} = 1.8 \pm 0.3$ pN) compared with the wild-type bond ($f_\beta^{WT} = 13.5 \pm 2.0$ pN),

without altering the elasticity-mediated force dependence of $C_{eff}$ (Fig. 5f and Supplementary Table 1). As a consequence, the R113G⁺/⁺ PCDH15 dimer remains bound to the CDH23 dimer 39% as long as the wild-type at zero-force but lasts just 0.3% as long at a resting tension of 10 pN. These results are also consistent with our biolayer interferometry measurements, which indicate a moderate reduction in a zero-force lifetime (Fig. 5e). Since cochlear hair cells experience considerably more force than vestibular hair cells or the calyceal processes of retinal photo-receptors[64], PCDH15–R113G⁺/⁺ likely unbinds more rapidly in the cochlea than in the vestibular system or retina, consistent with the more profound deafness phenotype in *PCDH15*–R113G⁺/⁺ human patients compared with the balance or visual phenotype.

**The lifetime of the tip link is insensitive to a broad range of physiological stimuli**. Individual tip links are subjected to a resting tension, which maintains transduction channels in their optimally responsive state[12,17,30,65]. Estimates for the operating range of a single mechanotransduction complex, characterized as the range of forces that change channel open probability $P_{open}$ from ~10% to ~90%, vary by three-fold or more depending on the organism, type of hair cell, and stimulus method[53,66–71]. However, sensitive experiments in bullfrog saccular hair cells suggest that channels open with a force of ~5 pN added to the resting tension (~10 pN), for an operating range of 10–15 pN[53,71]. These values are consistent with measurements of hair bundle movements in excised guinea pig cochlea preparations, where a sound pressure level of 60 dB moves a hair bundle <10 nm[72].

Our force spectroscopy experiments were performed with fixed loading rates and used to calculate the mean lifetime under static force (Fig. 3). However, in vivo, auditory stimuli in vertebrate hearing organs oscillate hair bundles with a sinusoidal force waveform[73] at speeds that are beyond the capability of our optical tweezer instrument. To understand the behavior of the tip-link connection in oscillating hair bundles in relation to the force operating range of the transduction channel, we performed Monte Carlo simulations of tip-link lifetime (Fig. 6) using the kinetic parameters calculated from linear loading (Supplementary Table 1) and an established quantitative model of hair bundle mechanics based on bullfrog saccular hair cells (Fig. 6a, b and see "Methods" section)[65]. Remarkably, we find that within the normal operating range of the transduction complex (~0–20 pN; corresponding to peak stereocilia displacements up to ~130 nm), the mean lifetime of the connection is essentially insensitive to changes in stimulus frequency (~20 Hz–10 kHz; Fig. 6d). Although forces during the peak half-cycle of the sine wave should increase the unbinding rate, there is also more efficient rebinding from state B1 → B2 during the slack half-cycle (Fig. 6c and Supplemental Movie 1). As the relative time spent at high force and low-force portions of a cycle is independent of frequency, lifetime is expected to be independent as well, provided the probability of rupture in an individual cycle is small. At very large stimulus magnitudes, corresponding to damaging noise stimuli[74], the lifetime is significantly reduced.

The forces acting on an individual tip link are also modified by the "slow-adaptation" myosin motors that adjust resting tension in tens of milliseconds[15,16,65,75]. To understand the effects of slow adaptation, we incorporated a motor component to our model, where the motor climbing rate is 1.6 μm s⁻¹ and the motor slip rate is 0.01 s⁻¹ with a slip distance proportional to force, yielding a resting tension of 10 pN (see "Methods" section). At low frequencies (when the period is longer than the adaptation time constant; here below ~10 Hz), adaptation acts to reduce the stimulus amplitude by reducing tension on the tip link within one stimulus cycle, preventing a high static load and largely negating

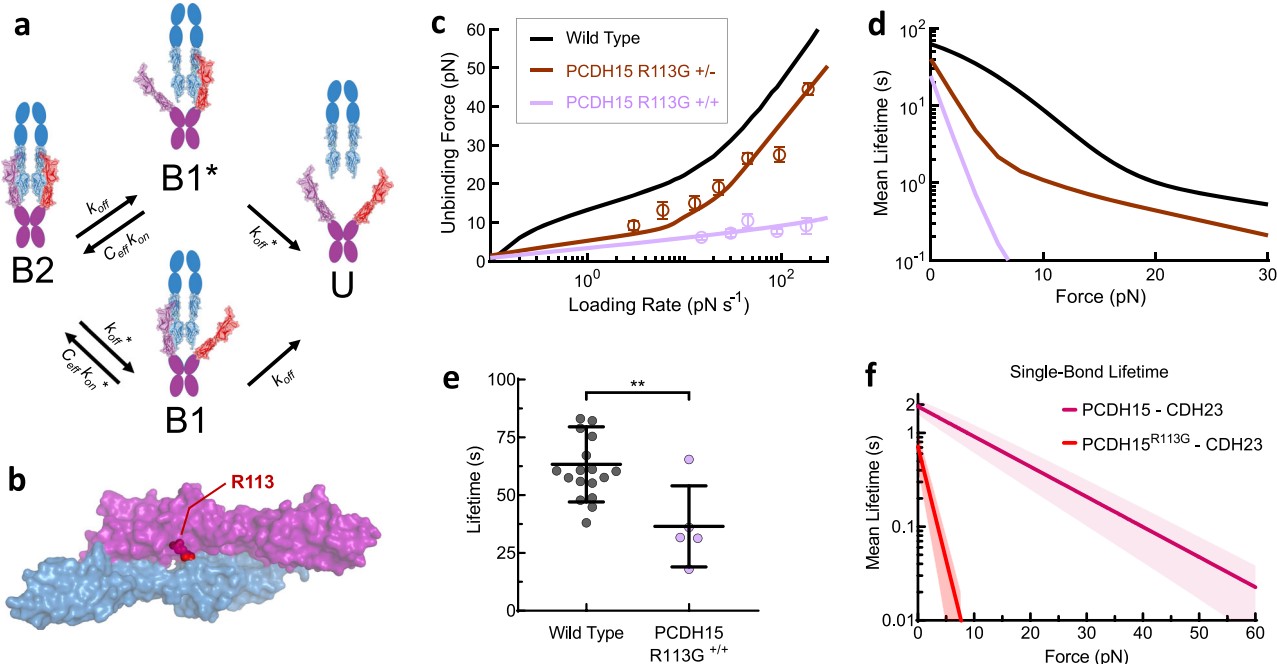

**Fig. 5 The mechanical phenotype of a hereditary deafness mutation. a** A kinetic state diagram representing a tip-link connection heterozygous for the R113G mutation (red) in PCDH15 (PCDH15 R113G$^{+/-}$). The doubly bound complex (B2) can transition to a single-bound state containing a mutated PCDH15 (B1*) or to a single-bound state with the wild-type PCDH15 (B1), with rates that differ between mutant and wild-type. **b** The location of the human deafness mutation R113G in the mouse PCDH15–CDH23 bond interface (PDB 4AXW). **c** Unbinding forces from dimeric, full-ectodomain proteins measured at 2 mM Ca$^{2+}$, with one (brown, $n = 316$ unbinding events) or both (purple, $n = 264$ unbinding events) PCDH15 strands mutated, simultaneously fit with the three- and four-state models. Error is shown as the optimal bandwidth from a kernel density estimation. **d** Calculated mean lifetimes of the wild-type and mutant tip-link connections. **e** Scatter plot of zero-force tip-link lifetimes measured from biolayer interferometry experiments with PCDH15 R113G$^{+/+}$ proteins (mean ± SD). Wild-type lifetime = 64 ± 16 s ($n = 18$ independent experiments), R113G$^{+/+}$ lifetime = 37 ± 18 s ($n = 5$ independent experiments). The Student's two-tailed unpaired $t$-test was used to determine statistical significance (**$p = 0.0041$). **f** Mean CDH23–PCDH15 single-bond lifetime as a function of force, calculated from single-bond kinetics of the PCDH15$^{R113G}$–CDH23 bond with parameters extracted from simultaneous fits to PCDH15 R113G$^{+/+}$ and PCDH15 R113G$^{+/-}$ force spectroscopy data (**c**). Shaded bands are the propagated error of the fit parameters $f_\beta$ and $k_{off}^0$. Relative to the wild-type tip-link bond, the R113G mutation increased both the zero-force single-bond off rate (from 0.5 to 1.4 s$^{-1}$) and the force sensitivity $f_\beta$ of the bond (from 13.5 to 1.8 pN). At a resting tension of 10 pN, a single wild-type bond lasts ~1 s, while a single PCDH15$^{R113G}$–CDH23 bond lasts <10 ms.

the effect of a prolonged high tension stimulus (Fig. 6e). This effect is likely most important in vestibular hair cells, which experience relatively slow, static deflections[76]. At higher frequencies, adaptation reduces resting tension because the slipping rate is faster than the climbing rate, so an oscillatory stimulus causes net relaxation[30]. With adaptation, even at a peak stimulus deflection of 1000 nm (peak force 72 pN above rest) the lifetime only dropped to 60% of that at resting tension (Fig. 6e). Overall, these surprising results indicate that multiple effects act to maintain an essentially constant lifetime of the connection at physiologically relevant frequencies and stimulus amplitudes, yet facilitate rupture of the connection when it is presented with stimulus magnitudes associated with damaging sound levels.

## Discussion

Our perception of sound and head movement is mediated by a mechanotransduction complex that converts nanometer-scale deflections into electrical signals. Many proteins of the complex have been identified, and how they assemble is beginning to be known. Yet, how this apparatus functions biophysically at the level of individual molecules is poorly understood. Our results characterize the strength and dynamics of the tip-link complex with single-molecule resolution and show how avidity governs the lifetime of the connection in response to force and concentrations of endolymphatic Ca$^{2+}$.

Although previous studies have depicted the tip link as a relatively static structure[77–79], our results indicate the tip link is a highly dynamic connection. Tip links ruptured by chelation of extracellular Ca$^{2+}$ or by exposure to damaging noise stimuli in vivo recover with a time constant of approximately 3–6 h[77,79], which suggested that normal tip-link lifetime should be at least as long as that recovery time. In contrast, we measure a tip-link lifetime of ~8 s at resting tension (Fig. 3b) and under physiological oscillatory forces (Fig. 6d, e), three orders of magnitude shorter than previously suggested. We speculate that a combination of off-target effects confounds the interpretation of recovery experiments. First, extracellular Ca$^{2+}$ chelators have detrimental effects beyond tip-link rupture, including disassembly of the actin core within shorter rows of stereocilia and removal of most lateral links[77,79,80]. These effects may slow the timecourse of recovery independent of intrinsic tip-link kinetics by decreasing the concentration of tip-link binding domains across the inter-stereocilia gap. Second, Ca$^{2+}$ chelation paired with high force lowers the energy barrier to tip-link protein unfolding[51,58], which may severely damage tip-link proteins and require lengthy refolding before reforming. Finally, treatment with Ca$^{2+}$ chelators abolishes CDH23 immunoreactivity at the tips of bullfrog stereocilia[12] and mostly removes CDH23 from the tip-link region in mouse[79], indicating either CDH23 recycling away from the site of mechanotransduction or unfolding of the epitope sites. Taken together, these effects make estimations of tip-link lifetime from recovery experiments difficult.

Our calculation of tip link single-bond kinetics at zero-force matches well with previously published results[11,26,46], and

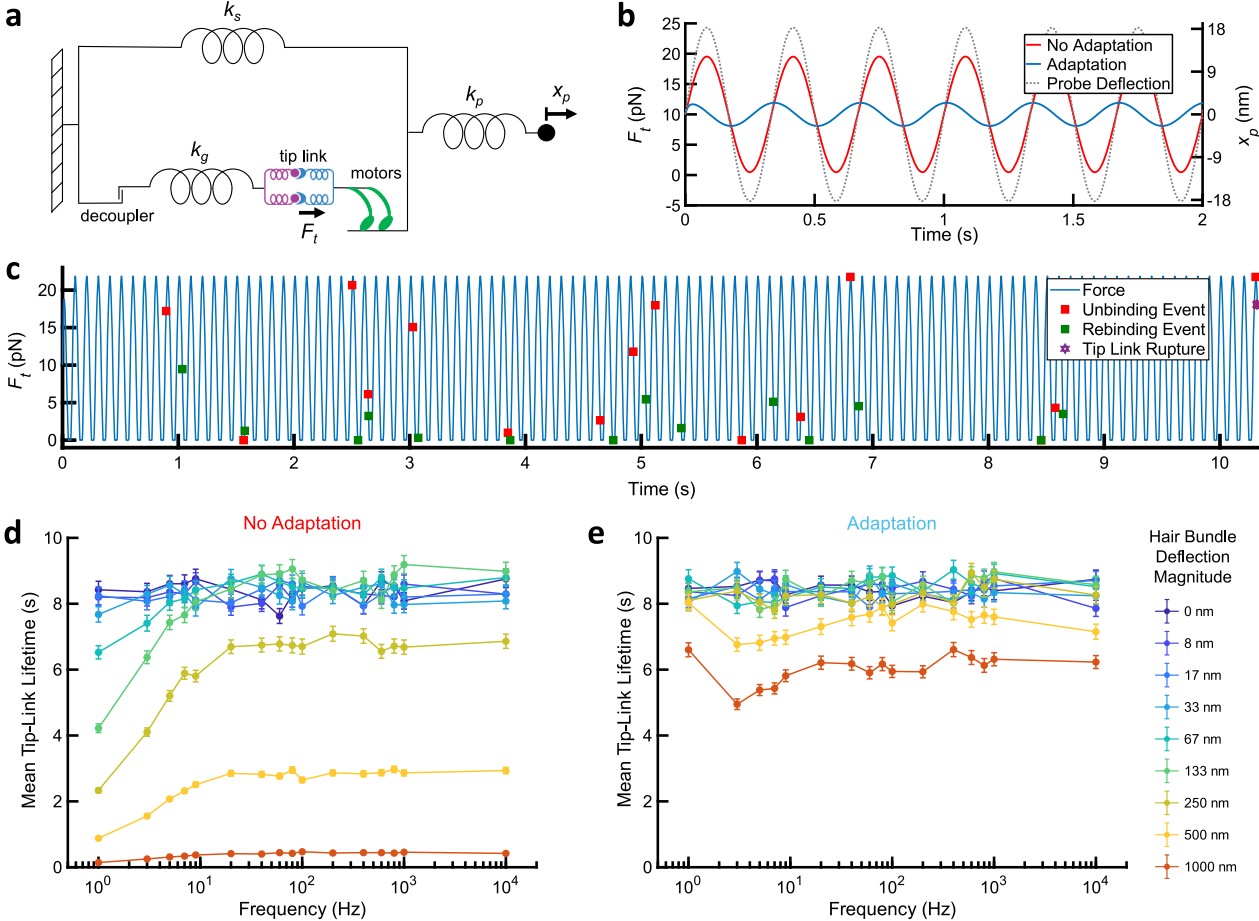

**Fig. 6 The lifetime of the tip link is insensitive to a broad range of physiological stimuli. a** A schematic of the modified motor-based adaptation model[65] (see "Methods" section) used to calculate mean tip-link lifetimes. $k_s$ is the pivot stiffness of the stereocilia bundle and $k_g$ is the gating spring stiffness. If a compliant probe is used to stimulate a hair bundle, $x_p$ is the probe deflection magnitude and $k_p$ is the probe stiffness (all normalized to a single tip link using a geometric factor $\gamma = 0.12$)[68,71]. The force on the tip link $F_t$ is then calculated. **b** Force on an individual tip link $F_t$ as a function of time, at a 3 Hz stimulus frequency, with (blue) and without (red) "slow" adaptation ($x_p = 18$ nm). The axis for probe deflection $x_p$ is at right. **c** A single Monte Carlo simulation trajectory of tip-link lifetime performed using force-dependent off-rates and concentration-dependent on-rates obtained from force spectroscopy of full-length dimers (Fig. 3), incorporating the effect of adaptation. Unbinding (red) and rebinding events (green) are marked with squares. An animation of the simulation trajectory is in Supplemental Movie 1. **d** Mean tip-link lifetimes, based on thousands of such simulations at frequencies between 1 and 10,000 Hz and at hair bundle deflection magnitudes of 0–1000 nm (error bars = SEM, $n = 5000$ simulations per data point). Without adaptation, tip-link lifetime is insensitive to both frequency and amplitude within the physiological range of hearing in humans (~20–10,000 Hz) for hair bundle deflections up to 133 nm, corresponding to a peak force of ~9 pN above resting tension on the tip link. **e** Mean tip-link lifetimes, with motor-based adaptation (error bars = SEM, $n = 5000$ simulations per data point). Slippage of the adaptation motors increases mean tip-link lifetime by reducing the force with each oscillation at low frequencies, and by reducing the resting force at high frequencies. With adaptation, tip-link lifetime is mostly insensitive to both frequency and amplitude for frequencies of 1–10,000 Hz and deflections to 500 nm.

appears to be independent of whether tip-link protein was expressed in bacteria and re-folded—which results in no post-translational modifications—or expressed in mammalian HEK293 cells where proteins are glycosylated and modified in the secretory pathway. However, post-translational modifications specific to the hair cell could act to change the affinity of the bond interface or to stabilize the PCDH15–CDH23 interaction, analogous to how phosphorylation of the C-terminal tail of most GPCRs increases their affinity for β-arrestins[81]. Unidentified accessory proteins may also act to stabilize the tip-link interaction, though high-resolution ultrastructures of the tip-link filament in hair cells depict a smooth strand without bulges of accessory proteins[5]. Finally, a subset of electron microscopy images from an early study of tip-link ultrastructure suggested the possibility of three tip link strands, based upon indents at the lower end of tip links after BAPTA chelation[5], which may

modestly extend the lifetime of the tip link. However, further biochemical studies have shown that full-length CDH23 and PCDH15 preferentially form dimers in vitro[10], both by themselves and when bound together. Interestingly, although these studies show that tip-link filaments form a helical structure, recent molecular dynamics simulations show that helix unwinding of full-length dimeric PCDH15 bound to CDH23 EC1-2 during stretching does not significantly twist the PCDH15 EC1-2 binding domains, likely due to strong *cis*-dimerization at EC3[52].

If the tip-link connection is ~8 s at both resting tension and under physiologically relevant sinusoidal stimuli, a fully unbound tip link must rebind, or new tip links reform, also on a timescale of seconds. The process would involve a PCDH15 dimer on one stereocilium binding to a CDH23 dimer on an adjacent stereocilium, presumably near the region where stereocilia touch. How can the free ends of an unbound tip link reform if all of the bound

tip links in a hair bundle are under a resting tension? Acute application of $Ca^{2+}$ chelators results in a mechanical deflection of even a quiescent hair bundle by 100 nm or more, leading to separation of stereocilia tips by ~10 nm or more[19]. If the stereocilia pair containing an unbound tip link moves as far as these experiments suggest, the two ends would likely be too far apart to reform. However, this motion is almost certainly a consequence of all of the tip links in a hair bundle breaking at once, given the correlative abolishment of transduction. Since the hair bundle is mechanically coherent[82], if only one tip link breaks, the remaining links on other stereocilia pairs should restrain the motion of stereocilia tips and hold the tip link ends close to each other so that they might reform quickly. Although tip links will be extended at rest by at least a few nanometers due to their intrinsic elasticity, making the possibility of the ends snapping away from each other after unbinding more likely, random thermal fluctuations in the stereocilia and in the tip links themselves may bring them close enough to reform. These concepts are consistent with recent mechanical observations of the bullfrog and rat hair bundles: the reformation of tip links occurs rapidly under select conditions after gentle $Ca^{2+}$ chelation, and reformation appears to be enhanced by negative deflection of the hair bundle[83].

For the reforming rate to match the ~8 s lifetime (Figs. 3b and 6e), where tip links have a 50% equilibrium bound probability, we can use the calculated on-rate of $3.5 \pm 0.7 \times 10^5$ $M^{-1}$ $s^{-1}$ to estimate that the effective concentration of free CDH23 and PCDH15 dimer ends must be at least ~0.3 μM. This concentration roughly corresponds to the two binding domains occupying a sphere with a diameter of less than ~210 nm. This diameter is ~30% longer than the length of an intact tip link and far longer than the 10–20 nm distance between stereocilia where they touch near their tips[5,84,85]. However, the concentration of free tip-link dimer ends is likely much higher than ~0.3 μM. From a structural point of view, CDH23 and PCDH15 are relatively stiff at endolymphatic $Ca^{2+}$ concentration[5,52,58], are complexed with membrane and intracellular proteins[6,43,86], and their insertion points appear to be spatially constrained even when broken[14,77,79]. These observations make it likely that free dimer ends occupy a much smaller space than a 210 nm sphere and are consequently at higher than 0.3 μM concentration. From the single-channel conductance of transduction channels and the number of stereocilia, it has been estimated that ~95% of channels in cochlear hair cells transduce during a stimulus[87]. If the lifetime of the tip link is ~8 s at resting tension, a corresponding ~95% tip link bound probability would necessitate a reforming rate of just hundreds of milliseconds and a local concentration of ~10–15 μM.

Rapid turnover of tip links might be tested by adding free tip-link proteins to the medium bathing hair bundles: if tip links unbind in seconds, the free proteins should quench rebinding and the measured mechanical sensitivity should decrease in seconds. A similar experiment has been done but did not result in a substantial decrease in transduction current[78]. However, those experiments used a free tip-link protein concentration of ~300 nM, far lower than the minimum effective concentration of free CDH23 and PCDH15 dimer ends of ~10–15 μM likely to be present for rapid rebinding, so they could not test the lifetime by quenching. Further tip-link competition experiments at adequately high concentrations could be difficult due to the instability of CDH23 and PCDH15 proteins at high concentration. Based on the lifetime of seconds in in vitro experiments, and the highly plausible estimate for the concentration of free dimer ends, we propose that the tip link maintains a dynamic connection across the stereocilia gap, unbinding on the scale of tens of seconds and rapidly reforming such that the majority of tip links are bound during normal stimuli.

We have shown that the tip-link connection is dynamic, and its lifetime can be modulated through disruption of *cis*-dimerization

or by altering the mechanoelastic properties via changes in extracellular $Ca^{2+}$. Modeling indicates that the lifetime with sinusoidal stimuli is essentially equal to that under resting tension for stimulus amplitudes of up to ~70 nm bundle deflection, even without accounting for adaptation. There is thus an opportunity for the sensory epithelium to regulate the probability that a tip link is bound—and therefore to adjust overall mechanoelectric transduction magnitude—through the concentration of extracellular $Ca^{2+}$. Extracellular $Ca^{2+}$ appears to be enriched nearly tenfold within ~10 μm of the hair bundle compared to bulk cochlear endolymph through the action of the PMCA2 $Ca^{2+}$ pump[88,89]. Additionally, $Ca^{2+}$ may be sequestered by the overlying tectorial membrane at rest and depleted during a high magnitude stimulus[90], perhaps through $Ca^{2+}$ entry through the mechanotransduction channel. Therefore, depletion of extracellular $Ca^{2+}$ near the hair bundle during a prolonged or high magnitude stimulus should change the elasticity of the tip-link complex, decreasing the lifetime of tip links and the proportion that are intact. This is consistent with reduced transduction observed following tectorial membrane $Ca^{2+}$ depletion[90]. Decreased $Ca^{2+}$ also has the effect of reducing the overall hydrodynamic diameter of each molecule[10], in part a consequence of the increased flexibility of EC-domain links in low $Ca^{2+}$ [11,58]. This will result in the free binding domains of each molecule being separated by a greater distance across the stereocilia gap, further reducing the probability the tip link will reform. This process is likely balanced through the re-establishment of local extracellular $Ca^{2+}$ through PMCA2 in the absence of transduction.

These mechanisms allow the tip link to act as a mechanical circuit breaker in two ways: the force-dependent unbinding rate becomes faster than the rebinding rate at high force, and the slight elasticity of the bound strand allows the free ends of the unbound cadherins to pull away from each other, decreasing the rebinding rate. This might be a protective mechanism, preventing large stimuli from tearing the transduction complex from the membrane.

Overall, our work reveals how nature uses force distributed across multiple bonds to create a tunable mechanical circuit breaker—a motif that is likely used in other natural systems and could be used as a design principle in synthetic systems.

## Methods

**Protein production and purification**. Using antibody Fc heterodimers[91,92], we engineered dimeric mouse PCDH15 and CDH23 proteins containing their first five EC domains, and we restricted binding to a single PCDH15–CDH23 bond by removing EC1-2 from one protein of each parallel dimer (Fig. 1b). This design was essential: monomeric EC1-2 and EC1-5 fusion proteins without the paired EC3-5 were unstable and bound non-specifically[11]. For single-bond tip link proteins with one EC1-2 binding domain, the coding regions of EC1-5 for mouse *Pcdh15* and *Cdh23* were amplified by PCR from full-length cDNA clones. This corresponds to base pairs 422-2269 from *Pcdh15* RefSeq cDNA NM_023115 and base pairs 408-2066 from *Cdh23* RefSeq cDNA NM_023370. Fusion proteins containing only the EC1-2 binding domains for each protein were unstable and bound non-specifically, and the addition of EC3-5 was necessary to stabilize these proteins, as determined from both biolayer interferometry and force spectroscopy. The EC3-5 domains were created by deleting amino acids 28–262 for PCDH15 and 25–228 for CDH23 using site-directed mutagenesis (New England Biolabs). This strategy preserved the native signal peptide sequence. The EC1-5/EC3-5 dimer architecture was necessary to eliminate the background binding of monomers when EC3-5 was exposed. Cleavage of the signal peptide was confirmed by transfecting and expressing individual plasmids and ensuring they were secreted from transfected Expi293 cells (ThermoFisher) and by glycosylation-corrected molecular weight determination using SEC-MALS (Wyatt Technology).

The coding sequences of antibody Fc heterodimerization domains were amplified from the pFUSE-hIgG1-Fc1 plasmid (Invivogen) containing two charged mutations on each opposing domain[91,92]. The SNAPtag sequence was amplified from the pSNAPf vector (NEB). Each of these fragments was cloned into a pFUSE vector (Invivogen) using NEBuilder® HiFi DNA Assembly Master Mix (NEB) along with a tandem 6× polyhistidine tag separated by a GSG linker.

For tip link fusion proteins containing two EC1-5 dimeric binding domains, shown in Fig. 2a, the same assembly principle was applied as described above. For full-length dimeric tip link fusion proteins, the coding sequences for the entire extracellular domain up to the last hydrophilic amino acid before the transmembrane domain for PCDH15 and CDH23 were used. The PCDH15 R113G mutation was introduced using site-directed mutagenesis (NEB). All plasmids encoding these proteins were screened for sequence fidelity by Sanger sequencing (GENEWIZ), and primers used to amplify these sequences are provided in Supplementary Table 2. DNA sequences were analyzed using SnapGene (Version 5.0).

Plasmids encoding each half of the final dimerized proteins were co-transfected into HEK Expi293 suspension culture cells in Expi293 medium at a ratio of 4:1 PEI: DNA (w/w), diluted in OptiMem (Invitrogen). A typical transfection used 15 µg of each plasmid with 120 µg PEI in 30 mL of cells at a confluency of $4 \times 10^5$ cells mL$^{-1}$. Cells were grown in 8% $CO_2$, shaking at 125 RPM. Transfected cells were grown at 37 °C for 16–20 h, then 7 mL Expi293 medium containing 2 mM sodium butyrate and 0.5× penicillin/streptomycin was added, and the cells moved to a 30 °C incubator. Cell supernatant was harvested 72–96 h post-transfection. Cells were removed by centrifugation at $7000 \times g$ for 15 min, and the remaining supernatant was 0.22 µm filtered and stored at 4 °C until purification.

Proteins were purified using Ni-NTA or TALON metal affinity purification. First, the supernatant was dialyzed against tris-buffered saline (TBS) containing 2 mM $Ca^{2+}$ (20 mM Tris, 125 mM NaCl, 2 mM $CaCl_2$, pH 7.4) overnight at 4 °C in 3.5 kDa MWCO dialysis tubing (Repligen). 1–1.5 mL of $Ni^{2+}$ Sepharose Excel (GE Life Sciences) or TALON resin (Takara) per 30 mL of supernatant was sedimented in a 10 mL purification column and then washed with 3 resin bed volumes of PBS followed by 0.5 bed volumes of TBS + $Ca^{2+}$ buffer. The filtered supernatant was next passed through the column by gravity at a rate of ~1–2 mL min$^{-1}$. For Ni-NTA purification, the column was then washed with 3 column volumes TBS-Tween+$Ca^{2+}$ buffer with 15–30 mM imidazole. For TALON purification, the resin was resuspended and washed with 15 column volumes TBS + $Ca^{2+}$ in batch mode three times. In each case, the protein was eluted by resuspending the $Ni^{2+}$ Sepharose with 3 mL TBS + $Ca^{2+}$ buffer with 200 mM imidazole and incubating in a tube on a rotator for 1 h at 4 °C. This mixture was then filtered through a 0.2 µM filter and concentrated on a 10 kDa (for EC1-5/EC3-5 proteins) or a 100 kDa (for full-length proteins) MWCO column (Vivaspin, GE Healthcare). The protein was buffer exchanged 2-3x using either 7 kDa MWCO Zeba Columns for EC1-5/EC3-5 proteins or 40 kDa MWCO Zeba Columns for full-length proteins (ThermoFisher).

Proteins were purified by size-exclusion chromatography on a Superose 6 Increase 10/300 column (GE Healthcare). Purity and molecular weight were confirmed by staining with a SNAPtag-reactive benzylguanine-Alexa 647 dye (NEB) and running the proteins on a 4–20% NuPAGE Tris-Glycine gel. The gels were stained with Krypton protein stain (ThermoFisher) and imaged using a laser gel scanner (GE Typhoon) at 532 and 635 nm excitation.

**DNA tethers.** Circular ssDNA from the M13 bacteriophage (New England Biolabs) was linearized at a single site using the restriction enzyme BtscI (New England Biolabs) and a site-specific oligonucleotide[93]. A forward primer containing dual 5′ biotins (Integrated DNA Technologies) was purchased and a reverse 5′ benzyl-guanine primer was synthesized from a primer with a 5′ primary amine on a 12-carbon linker (IDT) (Supplementary Table 2). The synthesis was accomplished with excess BG-GLA-NHS (NEB) in PBS pH 7.4 for 1 h at room temperature and buffer exchanged on a Zeba 7k MWCO desalting column. The forward and reverse primers were used to amplify a 2385 base pair DNA fragment from the linearized M13 template using Q5 DNA Polymerase 2× HotStart Master Mix (NEB). The amplified DNA tether was purified using Ampure XP beads at a ratio of 0.7:1 suspended beads to PCR mix and eluted in TBS + $Ca^{2+}$ buffer. To minimize damage to the DNA tethers, appropriate techniques were used to minimize unnecessary hydrodynamic stresses, and DNAse-free lab supplies were used to minimize potential degradation. The quality of the DNA tethers was checked with gel electrophoresis, and the tethers were stored at −20 °C in aliquots until use.

**DNA–protein tethering complex.** Purified SNAP-tagged tip-link fusion proteins (>1 µM) were reacted with >500 nM purified DNA tethers at room temperature for 1.5 h. Coupling efficiency was assessed by running the DNA–protein complexes on an agarose gel and staining for DNA (Supplementary Fig. 1d). The DNA–protein complex was desalted on Zeba Spin Desalting Columns (ThermoFisher) to remove excess imidazole (if present) and was then purified by Ni-NTA purification using Ni-Sepharose High-Performance resin (GE Healthcare) to remove free DNA tethers. The DNA–protein complex was purified as for protein purification, but the washing steps were carried out in the absence of imidazole and was performed entirely in batch mode. The resulting complex was concentrated to 20 µL on a 10–30 kDa MWCO column (VivaSpin, GE Healthcare) and buffer exchanged using Zeba Spin Desalting Columns (ThermoFisher).

**Bead passivation and functionalization.** Glass beads (silica microspheres of 3 µm diameter, 200 µL of 1% w/v; Bangs Laboratories) were cleaned and hydroxylated by first washing them in a glass tube in MilliQ water, then cleaning with 1% Hellmanex III detergent (Hellma Analytics) by boiling. Beads were then sonicated in

the detergent for 5 min, then washed and sonicated consecutively with acetone, 1 M KOH, and MilliQ water, and were then rinsed in anhydrous methanol.

The cleaned beads were aminosilanized by resuspending them in methanol and 20% v/v glacial acetic acid, then adding 1% v/v APTES (Sigma) in a glass tube and mixed by swirling. The solution was then covered with argon gas and rotated end-over-end for 10 min at room temperature. The beads were then rinsed repeatedly with methanol and MilliQ water to remove free aminosilane.

The beads were passivated with polyethylene glycol (PEG) using a variant type of "cloud point" PEGylation[94] (Supplementary Fig. 1a, b). 10 mg of mPEG-5k-NHS and 10 mg of Biotin-PEG-5k-NHS were dissolved in 86.2 µL of 0.5 M $Na_2SO_4$ in separate tubes. These reagents were mixed in varying ratios between 1:1 and 1:640 to a final volume of 86.2 µL, giving a cloudy solution. The varying ratios resulted in different surface biotin densities, used later to tune the density of DNA–protein tether on the surface of the beads. 13 µL of 1-M $NaHCO_3$ was added to the mixed PEG solution until the solution just became clear. The cleaned beads were pelleted using centrifugation, the supernatant removed, the PEGylation solution added to the beads, and the beads mixed into the solution by carefully pipetting without forming bubbles. The beads were then mixed on an end-over-end rotator with the top end of the tube at a 90° angle from the axis of rotation for 2 h. The beads were then washed with TBS + $Ca^{2+}$ buffer with 0.02% Tween-20 via successive centrifugation, bubbled with argon, and stored at 4 °C.

The surfaces of the beads were functionalized with streptavidin to enable the binding of biotinylated DNA tethers. 1 µL of PEGylated beads containing surface biotin were diluted 1:50 with TBS + $Ca^{2+}$ buffer + 0.02% Tween-20 + 0.1 mg mL$^{-1}$ Roche Blocking Reagent (TBS-TB + $Ca^{2+}$ buffer), and then 50 µM streptavidin was added and allowed to coat the surface for 10 min at room temperature while rotating. One batch of beads was coated with NHS-Alexa 594 labeled streptavidin to enable identification under the microscope via fluorescence and to distinguish PCDH15-linked beads from CDH23-linked beads. To remove free streptavidin, the beads were then washed five times by consecutive pelleting at $300 \times g$ for 1 min and washing with 200 µL TBS-TB + $Ca^{2+}$ buffer.

The beads were coated with the DNA–protein tethers by resuspending pelleted beads with 3–5 µL of the concentrated, His-purified, DNA–protein complex. The beads were rotated for 1 hr at room temperature and then washed three times with 200 µL TBS-TB + $Ca^{2+}$ buffer and resuspended in 10 µL of TBS-TB + $Ca^{2+}$ buffer. The coated beads were rotated at 4 °C until use.

For full-length tip link fusion proteins under some conditions (10 µM $Ca^{2+}$, R113G mutations, fast loading rates), it was necessary to perform experiments without the DNA tether in order to increase the rate of data collection. For these experiments, fusion proteins were directly biotinylated with BG-Biotin (NEB), buffer exchanged with a Zeba Desalting Column (ThermoFisher) three times, and directly coated onto streptavidin-coated beads. Individual tethers were verified using a WLC fit to the force-extension traces[95], and tether frequency was kept below 1:10 tethers:attempts. For the 95.62 pN s$^{-1}$ loading rate for full-length dimer fusion proteins in Fig. 3a, data were acquired with and without DNA tethers to confirm that the DNA tethers were not a source of mechanical instability in our measurements.

**Recording chamber preparation.** Plain 75 × 25 mm glass microscope slides (VWR) and 22 × 22 mm glass coverslips (Gold Seal, ThermoFisher) were cleaned by boiling in 1% Hellmanex III (Hellma Analytics) in MilliQ water. The glass was then sonicated for 30 min at 37 °C. After washing thoroughly in MilliQ water, the slides were dried and stored at room temperature in a sealed and desiccated container.

A recording chamber was prepared by laying thin strips of Kapton Tape (DuPont) onto a pre-cleaned microscope slide, then placing and sealing the glass coverslip on top by applying pressure along the tape lines. This created several channels with a height of ~1 mm. Each channel was flushed with 5 mg mL$^{-1}$ Roche Blocking reagent dissolved in PBS and allowed to sit in a humidified chamber for 10–30 min at room temperature in order to block the glass surfaces and prevent bead sticking. The channel was then flushed with 10–20 chamber volumes of TBS-TB + $Ca^{2+}$ buffer. Each chamber was used immediately after blocking and washing.

1 µL each of the coated beads containing either PCDH15 or CDH23 protein-DNA tethers was diluted 1:100 in TBS-TB + $Ca^{2+}$ buffer and injected into the recording chamber. The chamber was then hermetically sealed with vacuum grease and locked onto the microscope stage of the instrument. For each condition, measurements were taken from experimental and sample replicates.

**Instrument description.** The optical tweezers system used in these experiments is functionally similar to the one previously described[96], with several important differences. First, the micropipette was replaced with a second optical trap. This was achieved by using a polarizing beam splitter to split the 1064 nm laser beam into two polarized components. One beam remained stationary as in the previous setup, and the second beam was driven by a piezo-mirror with fine control (Mad City Labs, Madison WI), steered in the back focal plane of the objective.

The coverslip containing the recording chambers was mounted on an M-686 XY Stage with Piezoceramic Linear Motors (Physik Instrumente) which was directly attached to the microscope.

**Single-molecule experiments.** Experiments in 2 mM $Ca^{2+}$ were performed at room temperature in TBS-TB + $Ca^{2+}$ buffer. For experiments in $Ca^{2+}$ concentrations below 2 mM, TBS-TB + $Ca^{2+}$ buffer was made without $Ca^{2+}$ or Tween-20 (20 mM Tris, 125 mM NaCl, pH 7.40) using an established method[58]. Residual $Ca^{2+}$ from this solution was removed by passing 100 mL of TBS buffer over 5 g of pH-neutralized Chelex 100 resin (BioRad) in a column three times at a flow rate of ~3 mL min$^{-1}$, which yields a solution containing <0.2 μM $Ca^{2+}$ [58]. $Ca^{2+}$ was added back to the $Ca^{2+}$-free TBS buffer to the desired concentration from a stock standard of 1 M $CaCl_2$ solution (Sigma).

Beads were identified on the optical tweezer microscope by Alexa 594 fluorescence and picked up in each of the trapping beams. Custom software (LabVIEW) was used to control data acquisition and piezo motion. Bead positions were determined in real time at 1400 frames per second with sub-pixel fitting of a polynomial to the dark edges of the beads. The stiffness of each trap was calibrated at 50 mW of laser power using a blur-corrected power spectrum fit[97]. Experiments were performed at 950 mW of laser power, which corresponded to a trap stiffness of ~0.1 pN nm$^{-1}$.

Experiments were performed by bringing the piezo-mirror-driven bead close to the stationary trapped bead for 0.05–5 s and then retracting at a constant velocity. The formation of an individual tether was cursorily checked by tether length and by displacement of the stationary bead from the center of the trapping beam (Supplementary Fig. 1f). When the software detected a single tether, the bead was brought back to the original bead, just close enough that the tether was slack but that there was no chance of forming a second tether. The tether was then quickly pulled out at a constant velocity until the tether ruptured completely. Bead retraction was performed at force-loading rates between 0.39 and 377.8 pN s$^{-1}$.

**Data analysis.** Tethering and unbinding events were analyzed individually using custom software (MATLAB). For each set of beads, the average diameter of the two beads was calculated using the polynomial fits to the edges of the beads obtained from the imaging camera. For bead-diameter determination, each set of beads was held stationary in the optical traps for 1400 frames and the diameter of each bead in pixels was determined using the averaged distances from polynomial fits. Pixel distances were converted to nm using a conversion factor of 30.3162 nm pixel$^{-1}$ at the defined magnification and verified using a reticule. The extension was measured as the edge-to-edge distance between the two beads in each frame. Individual unbinding traces were fitted with a worm-like chain model to obtain unbinding force, contour length, and persistence length. The distributions of contour lengths and persistence were verified (Supplementary Fig. 1g) to ensure only single-molecule bonds were analyzed. Force-loading rates were calculated from the linear force regime of each unbinding trace, and the average loading rate at each force was calculated (Supplementary Fig. 1h).

Unbinding forces from at least three experimental replicates were averaged together for each loading rate tested, to account for preparation variability (see Supplementary Table 1 for rupture data statistics). The most-probable unbinding force is the force where the unbinding probability density is at a maximum. For each set of unbinding forces at a given loading rate we used a kernel smoothing function (ksdensity in MATLAB with default options) to estimate the unbinding probability density as a function of force[98–100]. The error estimate for each most-probable unbinding force was taken to be the kernel bandwidth, which was systematically chosen based on the spread of the unbinding force data and the number of rupture forces measured.

Single-bond unbinding data were analyzed using an Evans–Ritchie model for the dynamic strength of molecular adhesion bonds[39]:

$$f^* = f_\beta \cdot \ln\frac{r_1}{f_\beta k_{off}^0} \quad (1)$$

where $f^*$ is the most-probable rupture force, $f_\beta$ is the force scale, $r_1$ is the force-loading rate, and $k_{off}^0$ is the off rate at zero-force. The force scale $f_\beta$ is defined as:

$$f_\beta = \frac{k_B T}{x_{ts}} \quad (2)$$

where $k_B$ is the Boltzmann constant, T is absolute temperature, and $x_{ts}$ is the distance to the transition state for unbinding. Double-bond unbinding data were fit with a multi-state avidity force-dependent avidity model in MATLAB.

**A model for force-dependent avidity.** Conforming to the overall structure of the tip-link connection, we modeled the system as two bonds loaded in parallel. In this condition, the force applied to the tip link is distributed equally between the two-component bonds. If one bond breaks, the remaining bond assumes the entire force load unless and until the other bond rebinds. We, therefore, described the two-bond interaction using the following differential equations:

$$\frac{dB_2}{dt} = C_{eff} k_{on} B_1 - 2k_2 B_2 \quad (3)$$

$$\frac{dB_1}{dt} = -(C_{eff} k_{on} + k_1) B_1 + 2k_2 B_2 \quad (4)$$

where $B_2$ represents the fraction of tip links in the double-bound state, and $B_1$ represents the fraction of tip links in the single-bound state. $k_{on}$ is the solution on-

rate of a single PCDH15–CDH23 bond. $C_{eff}$ is the effective concentration of the unbound EC1-2 interface domains. $k_2$ is the force-dependent off-rate of the PCDH15–CDH23 bond when there are two bonds splitting the force load (B2 → B1):

$$k_2 = k_{off}^0 \cdot e^{\frac{F}{2f_\beta}} \quad (5)$$

where $k_{off}^0$ is the solution off-rate of a single PCDH15–CDH23 bond, $F$ is the applied force, and $f_\beta$ is the force, which accelerates the off-rate by e-fold. $k_1$ is the force-dependent off-rate of the PCDH15–CDH23 bond when there is one bond bearing the entire force (B1 → U):

$$k_1 = k_{off}^0 \cdot e^{\frac{F}{f_\beta}} \quad (6)$$

In the case where $F = 0$, these equations can be solved analytically. From this we calculate the mean lifetime $\tau$ to be:

$$\tau = \frac{C_{eff} k_{on} + 3k_{off}}{2k_{off}^2} \quad (7)$$

In these experiments, the force changes linearly in time with a loading rate $l_r$:

$$F = l_r t \quad (8)$$

We assume the effective concentration can change as force is applied due to the extensibility of the filaments. Each filament can be thought of as a series of springs each with a spring constant $\kappa$. If one strand breaks, the free ends separate, and tension on the bound strand is doubled, leading to a lower effective concentration $C_{eff}$.

$$C_{eff}(F) = C_{eff}^0 \cdot e^{-\left(\frac{F}{f_c}\right)^2} \quad (9)$$

$C_{eff}^0$ is the local effective concentration at zero-force; the reduction in $C_{eff}$ with force is characterized by a force[50]:

$$f_c = \sqrt{2\kappa k_b T} \quad (10)$$

The most-probable unbinding force of the dimeric interaction is calculated by finding the most-probable unbinding time and multiplying it by the loading rate. The most-probable unbinding time is calculated by finding the maximum of the unbinding probability density function:

$$-\left(\frac{dB_2}{dt} + \frac{dB_1}{dt}\right) = k_1 B_1 \quad (11)$$

The heterogeneous system is described by the following reaction equations:

$$\frac{dB_2}{dt} = C_2 k_{on[1]} B_{12} + C_1 k_{on[2]} B_{11} - 2(k_{21} + k_{22}) B_2 \quad (12)$$

$$\frac{dB_{12}}{dt} = -\left(C_2 k_{on[1]} + k_{12}\right) B_{12} + k_{21} B_2 \quad (13)$$

$$\frac{dB_{11}}{dt} = -\left(C_1 k_{on[2]} + k_{11}\right) B_{11} + k_{22} B_2 \quad (14)$$

where $B_2$ represents the fraction of tip links with two bonds. $B_{1j}$ represents the fraction of tip links with one bond where the index $j$ indicates which bond type. $k_{on[j]}$ is the solution on-rate of a single PCDH15–CDH23 bond. $k_{ij}$ is the force-dependent off-rates of the PCDH15–CDH23 bond, where the index $i$ denotes the number of bonds and $j$ denotes the bond type:

$$k_{ij} = k_{off[j]} e^{\frac{F}{if_\beta}} \quad (15)$$

$k_{off[j]}$ is the solution dependent off rate for the $j$th bond type. $C_j$ is the force-dependent concentration for the $j$th bond type.

$$C_j = C_o e^{-\left(\frac{F}{f_{c[j]}}\right)^2} \quad (16)$$

Here we assume that the effective concentration for the $j$th bond type decays as force is applied with a characteristic for $f_{c[j]}$.

The most-probable unbinding time of the dimeric heterogeneous interaction is calculated by finding the maximum of the unbinding probability density function:

$$-\left(\frac{dB_2}{dt} + \frac{dB_{12}}{dt} + \frac{dB_{11}}{dt}\right) = k_{11} B_{11} + k_{12} B_{12} \quad (17)$$

The most-probable unbinding time is multiplied by the loading rate to yield the most-probable unbinding force.

Each model was fit to the data using fminsearch in MATLAB to find the minimum of the $\chi^2$.

$$\chi^2 = \sum_{i=1}^{N} \frac{\left(F_i^{exp} - F_i^{model}\right)^2}{dF_i^2} \quad (18)$$

where $F_i^{exp}$ is the $i$th experimental most-probable unbinding force and $F_i^{model}$ is the most-probable unbinding force predicted by the model. $dF_i^2$ is the estimated square error for the $i$th unbinding force. To estimate the error for each fit parameter obtained, we calculated the local curvature of the $\chi^2$ as a function of that fit

parameter. The variance $(\sigma_{xi})^2$ of the fit parameter $x_i$ is estimated by:

$$(\sigma_{xi})^2 = 2\left(\frac{\partial \chi^2}{\partial x_i^2}\right)^{-1}. \qquad (19)$$

**Biolayer interferometry**. Proteins were produced and purified in the same manner as for force spectroscopy experiments. Protein molarity was verified using two complementary methods: UV absorbance at 280 nm in solution, and fluorescence intensity in an SDS-PAGE gel stained with the quantitative Krypton (Thermo-Fisher) protein stain relative to a BSA standard (ThermoFisher). To create passivated two-dimensional streptavidin-coated biosensors, we used Aminopropylsilane Dip and Read Biosensors (ForteBio), which contain a two-dimensional surface of primary amine groups. Dry sensors were first hydrated in a variant cloud-point buffer (100 mM HEPES, 0.6 M $K_2SO_4$, pH 7.75) in the sensor tray for at least 5 min. All experiments were performed in a 384-well tilted-bottom microplate (Pall ForteBio). Amine-reactive NHS-PEG$_{12}$-Biotin (ThermoFisher) was dissolved to 4 mM in cloud-point buffer and distilled water was added in ~5 μL drops until the buffer just became clarified. Sensors were immediately coated in NHS-PEG$_{12}$-Biotin solution for 2300 s on an Octet Red384 instrument (ForteBio) in real time to assess sensor coating efficiency. Sensors whose signals did not group together during the coating were discarded. Sensors were then repeatedly and sequentially washed in cloud-point buffer and ultrapure water, then equilibrated in TBS-Tween +$Ca^{2+}$ buffer. Biotinylated sensors were then coated for 200 s with 5 μM strep-tavidin (ProZyme) and washed twice in TBS-Tween+$Ca^{2+}$ buffer. Drift sensors were quenched with 10 μM biocytin (Sigma Aldrich).

In each experiment, CDH23 was biotinylated using a custom-synthesized benzylguanine-PEG15-biotin linker, excess linker was removed using a desalting column or polyhistidine purification, and the proteins were buffer exchanged into TBS-Tween+$Ca^{2+}$ buffer. CDH23 was immobilized to a binding signal of 0.175–0.25 nm above baseline in TBS-Tween+$Ca^{2+}$ buffer and washed in at least four separate wells of TBS-Tween+$Ca^{2+}$ buffer before association with PCDH15. Biocytin-quenched drift sensors were dipped into the same concentration of CDH23 for the same length of time as the experimental sensors, in parallel. To assess kinetics, the CDH23+ and CDH23− sensors were dipped into a baseline solution of TBS-Tween+$Ca^{2+}$ buffer, then dipped into a solution of 10–300 nM PCDH15 or PCDH15 lacking EC1-2 in TBS-Tween+$Ca^{2+}$ buffer to measure association, and finally into the same baseline solution to measure dissociation. PCDH15 and PCDH15 lacking EC1-2 proteins were kept equimolar during each experiment. Experiments were performed using both biological and experimental replicates. Solutions for low $Ca^{2+}$ experiments were prepared as described for single-molecule experiments.

PCDH15 proteins lacking EC1-2 were produced by deleting the EC1-2 coding region from DNA plasmids encoding the full-length extracellular fusion proteins using the same genetic deletion strategy employed for the EC1-5/EC3-5 dimers used for force spectroscopy. The change in apparent molecular weight was confirmed on an SDS-PAGE gel and via size-exclusion chromatography (Supplementary Fig. 2a).

Each experimental kinetic trace was first analyzed using the ForteBio Octet Data Analysis Software 10.0 (Pall ForteBio). Each sensor trace from the biocytin-quenched drift sensor was subtracted from the corresponding experimental sensor, and the subtracted traces were processed using Savitzky-Golay filtering. The drift-corrected sensor trace from the control PCDH15 lacking EC1-2 was then subtracted from the drift-corrected PCDH15 experimental sensor trace. The final subtracted traces were exported into Prism 8.1 software (GraphPad) and each dissociation phase was fitted using single-exponential kinetic equations. On-rates were measured at 300 nM PCDH15. Data were collected several times in experimental replicates (WT 2 mM $Ca^{2+}$: $n = 18$, 50 μM $Ca^{2+}$: $n = 9$, 30 μM $Ca^{2+}$: $n = 6$, 10 μM $Ca^{2+}$: $n = 9$; PCDH15 R113G$^{+/+}$ 2 mM $Ca^{2+}$: $n = 5$).

**Monte Carlo simulations of tip-link lifetime**. We used an established quantitative model of hair bundle mechanics[65] to calculate the force on a single tip link in response to an oscillatory bundle deflection, at frequencies from 1 Hz to 10 kHz. We modeled a hair bundle with a gating spring stiffness of 0.6 mN m$^{-1}$ and a geometry factor of $\gamma = 0.12$ (Fig. 6a, b); thus positive deflection by 133 nm would increase tip-link tension by 9 pN, enough to open essentially all channels if the gating swing is 4 nm[68,71]. We assumed a resting tension of 10 pN[53]. In this model, the gating spring does not exert compressive force, so a waveform with high amplitude cannot produce negative tension. For each δt of the simulation (1–10 μs depending on conditions), we calculated the tip-link tension, the force-dependent off-rates, the force-dependent $C_{eff}$ and the rebinding rate, and then the probability that the tip link remained bound. Each simulation was run until both strands unbound, typically 5–10 s. To obtain a histogram of lifetimes, we ran the Monte Carlo simulation 5000 times for each condition, with phase-randomized sinusoidal stimuli (Fig. 6d, e).

To model "slow adaptation", we used a motor climbing rate of 1.6 μm s$^{-1}$ and a motor slip rate of 0.01 s$^{-1}$, yielding a resting tension of 10 pN. When calculating the effect of adaptation, we calculated the slipping and climbing of the adaptation motor in each δt and how that affected tip-link tension.

**Quantification and statistical analysis**. All kinetic measurements using biolayer interferometry were measured at least five times in each experimental condition ($n = 5$–18). Experimental replicates were performed at least three times. We used ForteBio Octet Data Analysis Software 10.0 (Pall ForteBio) and Prism 8.1 software (GraphPad) for kinetic fitting. The Student's two-tailed unpaired $t$-test was used to determine statistical significance (*$p < 0.05$, **$p < 0.01$, ***$p < 0.001$).

**Reporting summary**. Further information on research design is available in the Nature Research Reporting Summary linked to this article.

## Data availability

Data supporting the findings of this manuscript are available from the corresponding authors upon reasonable request. A reporting summary for this Article is available as a Supplementary Information file. Source data are provided with this paper.

## Code availability

Code used to analyze data in this paper were performed in MATLAB, and are available at a dedicated Github repository [https://github.com/Corey-Lab/Mulhall-et-al-Nature-Communications].

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

## Acknowledgements

We thank Dr. Marcos Sotomayor for helpful discussion and early conceptualization, and members of the Corey and Wong Laboratories for helpful discussions and for critical reading of the manuscript. This work was supported by the NIH (F31 DC016199 to E.M.M., R01 DC000304 to D.P.C., R01 DC002281 to D.P.C. and W.P.W., and R35 GM119537 to W.P.W.). E.M.M. was a Harvard Medical School Department of Neurobiology Graduate Fellow.

## Author contributions

Conceptualization, E.M.M., A.W., M.A.K., D.P.C., and W.P.W.; formal analysis, E.M.M., A.W., W.P.W., and D.P.C.; funding acquisition, D.P.C. and W.P.W.; investigation, E.M.M., A.W., and M.A.K.; methodology, E.M.M., A.W., D.Y., D.P.C. and W.P.W.; software and instrumentation, A.W. and W.P.W; supervision, D.P.C. and W.P.W.; visualization, E.M.M., A.W., D.P.C., and W.P.W.; writing—original draft, E.M.M., A.W., D.P.C., and W.P.W.

## Competing interests

The authors declare no competing interests.
