## [Peer Review File · Nature Communications]

Reviewer #1 (Remarks to the Author):

The manuscript by Mulhall reports on technically superb experiments measuring the force sensitivity of the interaction of a PCDH15 dimer and a CDH23 dimer, which is thought to be the structure of the tip link in vertebrate hair cells. The authors methodology used by the authors to present dimers to each other and then measure bond lifetime in response to force stimuli is both clever and robust.

The authors show that the interaction of the PCDH15 and CDH23 dimers is qualitatively and quantitatively distinct from the interaction of monomers, showing the greater than additive increase in stability with the dimer-dimer interaction. The power of this paper is the combination of measurements of force-dependent monomer-monomer interaction, zero-force dimer-dimer interaction (the bilayer interferometry experiments), force-dependent dimer-dimer interaction, Ca²⁺ sensitivity, changes wrought by a deafness mutation, and modeling. These observations are all consistent with each other.

The paper presents a provocative suggestion, that the tip link is far more dynamic than we thought otherwise. If the tip-link cadherins behave in vivo as they do in vitro (e.g., no other biological processes modify interaction dynamics), is difficult to argue against the idea that the tip link is remarkably dynamic. That said, the paper would have been strengthened considerably by an in vitro demonstration that their suggestion is correct. The present manuscript is complete and strong, but I guess we will need to wait for future experiments to determine whether the model presented works in an actual hair bundle.

I otherwise have relatively few comments. The paper is beautifully written and illustrated, and the data appear to be extremely reliable, as different experiments nicely reinforce each other (and are consistent with many measurements reported in the literature).

Major comments

1. I have one major point that the authors should address more thoroughly, which is brought up (albeit from a different perspective) in the Discussion, page 16. The authors need to consider a wider range of past data in discussing the implications of the 8 sec tip link lifetime. The question is whether after tip-link dissociation, the PCDH15 dimer ends and the CDH23 dimer ends can possibly be close enough together to drive rebinding. One argument against that idea is that based on experiments like Assad et al. (1991), which show a large positive bundle movement after breaking all the tip links; if the stereocilia pair that formerly housed the tip link in question moves as far as the bundle movements measured, the two ends will likely be far too far apart. However, the large positive movement is a consequence of having all the tip links break at once; if only one breaks, the remaining stereocilia pairs with intact tip links will likely hold the stereocilia pair with a broken tip link in place, given the coherence of the bundle. (The authors could make this point.)

However, there is at least one more challenge. Those same experiments and others indicate that the tip link is extended at rest, and the rest extension has been calculated to be on the order of 10-15 nm. Moreover, there are data that suggest that the CDH23 insertion point climbs up somewhat after tip links break (Kazmierczak et al., 2007), which make the separation even greater. At some point the tip link ends waving in the breeze will never find each other as their bases are too far apart.

Based on what we know about the hair bundle, a tip-link lifetime of 8 sec under resting conditions

would suggest that starting with a bundle with intact tip links, the bundle would rapidly lose transduction as tip links break but are not re-engaged. The authors should explain more thoroughly why they expect that tip links will reform after breaking. (It is possible that new CDH23 dimers are recruited, but then why does the upper insertion point appear to be a stable structure? Maybe it isn't.)

2. The argument at the bottom of the page about the concentration of PCDH15 or CDH23 fragments necessary for inhibition of reformation is circular. The authors suggest that one needs to use at least $0.34 \mu\text{M}$ of fragments, as that is the calculated free-end concentration. But the calculation of the free-end concentration assumes that if tip links break rapidly (i.e., 8 sec lifetime), they reform rapidly.

The authors also dismiss the idea that the fragment-blocking experiment can be done because the proteins are unstable at high concentrations. Yet they describe experiments where they use $>1 \mu\text{M}$ tip-link fusion proteins in coupling reactions with DNA tethers for 1.5 hr. It seems like all the ingredients for doing the experiments are in the authors' hands—one would expect then that applying $>1 \mu\text{M}$ tip-link protein to a quiescent hair-cell prep should cause a rapid loss of transduction, with a half life on the order of seconds. It is a shame that the authors did not perform this experiment as it would have nailed their model for the dynamic nature of the tip-link bond.

Other comments

Page 2, first sentence of second paragraph. While it is true that the tip link has been shown by EM to be a double-stranded complex, given the far greater molecular information we have now as compared with the year 2000, I recommend modifying this description. More accurately it is an antiparallel structure of one PCDH15 dimer and one CDH23 dimer, and this first sentence should reflect that.

Page 5, line 5. The increase was sublinear, not “slower than linear” (what does that mean, actually?). Also, for this point, what data show this sublinear behavior? Point to them.

Page 5, line 10 and elsewhere. The SI unit of time is the second, and it is abbreviated “s”. Please change “ 1.9 ± 0.4 seconds” to “ 1.9 ± 0.4 s”. The inverse time unit is s^{-1} , which the authors use correctly.

Page 5, lines 11-12. Please explain how the f -beta measurement was extracted from the data in Fig. 1f.

Page 7, line 4. Please write “ 3.46 ± 0.70 ” or “ 3.5 ± 0.7 ”—it does not make sense that the precision of the mean is greater than the precision of the error.

Page 14, last sentence. It is a little funny to say bath applied or paired with a stimulus. It's like apples and giraffes. The chelators are usually bath applied or applied by iontophoresis, which are significantly different approaches, and they are either applied to quiescent bundles (usually true with bath application) or bundles undergoing mechanical stimuli (usually true with iontophoresis). But don't mix those different conditions—the naïve reader will likely be confused as to the meaning.

Page 15, second sentence. I am puzzled as to why Zhao et al. is cited for this concern (i.e., that

chelation coupled with force could lead to tip link unfolding). Zhao et al. applied chelators to quiescent bundles.

Page 16, last paragraph. 300 nM is the same (more or less) as 0.34 μM , not an order of magnitude lower. The Lelli et al. (2010) paper reports a half-maximal inhibitory concentration of CDH23 fragment of 40 nM, not 300 nM. Thus while the order of magnitude difference is correct, the concentration reported is not. That said, 40 nM is just the half-maximally effective concentration. The Lelli et al. paper is vague as to what conditions were used in any attempts to block transduction in the absence of chelator treatment—their Fig. 5B indicates that they used a concentration as high as 200 nM in the post-chelator experiments, at least.

Page 19, last line. Change to “2 mM (pink) Ca^{2+} .”

Page 21. Move the description of the biolayer interferometry method from 2a to 2c, where the experiment is described.

Page 22. First line of 3a legend. What does “These” refer to?

Signed, Peter Barr-Gillespie

Reviewer #2 (Remarks to the Author):

This manuscript carefully measures the dynamics and strength of Tip-link connection of single-bond, double-stranded, and hereditary deafness mutation cases under various extracellular Ca^{2+} , with single-molecule force spectroscopy (under force condition) as well as ensemble biolayer-interferometer method (under no force condition). With a derived kinetic model considering a force-dependent off-rate of the PCDH15-CDH23 bond and a force-dependent effective concentration C_{eff} , the manuscript calculates a relation, which can't be measure directly, between average life-time of bond and applied force. Furthermore, Monte Carlo simulation was conducted in this manuscript to understand the behavior of tip-link connection in oscillating hair bundles in relation to physiological force range. Overall, this manuscript presents a complete and sound research project with both single-molecule results, ensemble measurements, theory calculation and simulation; these results provide significant foundation for further studies in related field; hence the reviewer suggests to accept this manuscript to publish on Nature Communication with minor revision:

- 1) Rupture force of PCDH15-CDH23 bond is measured with a configuration showed on Fig. 1c by dual optical traps. How to the authors know the measured rupture force is resulting from the breaking of PCDH15-CDH23 bond, rather than from other bonds or tethers, i.e: poor DNA tethers breaks or unbinds from the beads? Or how do the authors confirm that the data of broken DNA tethers is not mixed into the data of rupture force from PCDH15-CDH23 bond?
- 2) While stretching protein-DNA tethers (Fig. 1c and Fig. 1d), how does PCDH15-CDH23 tip-link behave before rupture, is it similar as a spring? In case the tip-link was stretched for some distance in the experiments, how does it compare with physiological moves of hair bundle due to sound pressure?
- 3) In Fig. 1f, could you please explain and discuss more on X-intercept, where the unbinding force is 0pN ? How to relate this number with ensemble measurements?

- 4) Fig. 2c presents association signal as a function of time, which is measured from biolayer interferometry method. Does Fig. 2c show single-molecule data or averaged ensemble data? Does this signal result from $U \diamond B_2$, or $B_1 \diamond B_2$? If the signal is from $B_1 \diamond B_2$, how does this data been measured? In case the association signal is from $U \diamond B_2$, there should be a state conversion from $U \diamond B_2$ in Fig. 2b?
- 5) In Fig. 3a, is the green dashed line ($C_{eff} = \text{constant}$) indicating the local effective concentration at zero force C_{eff0} ? If yes, how to understand that black solid line ($C_{eff} = C_{eff}(F)$) is higher than green dashed line ($C_{eff} = \text{constant}$) in regions of low loading rate ($< 100 \text{ pN s}^{-1}$)? To my understanding, the $C_{eff} = C_{eff}(F)$ should not be higher than that of C_{eff0}
- 6) In Fig.4, the reviewer suggests to plot Fig. 4a' and Fig. 4b' with the biolayer interferometry measurement, similar with Extended Data Fig.3, which is easier for reader to compare both measurements.
- 7) On page 36 and 37, the most probable unbinding force of the dimeric interaction is calculated by finding the most probable unbinding time and multiplying it by the loading rate. The unbinding force is a most probable rupture force. In Fig.3a, Fig.4a and Fig.4b, the measured unbinding force data fits very well with calculated most probable rupture force. Could the authors say more about this good fit?
- 8) In main figures, the x-tick and y-tick are not consistent (i.e.: Fig.4b and Fig.4c). Some are in and some are out. Please adjust ticks.

Response to Reviewer's Comments

Reviewer #1 (Remarks to the Author):

The manuscript by Mulhall reports on technically superb experiments measuring the force sensitivity of the interaction of a PCDH15 dimer and a CDH23 dimer, which is thought to be the structure of the tip link in vertebrate hair cells. The authors methodology used by the authors to present dimers to each other and then measure bond lifetime in response to force stimuli is both clever and robust.

The authors show that the interaction of the PCDH15 and CDH23 dimers is qualitatively and quantitatively distinct from the interaction of monomers, showing the greater than additive increase in stability with the dimer-dimer interaction. The power of this paper is the combination of measurements of force-dependent monomer-monomer interaction, zero-force dimer-dimer interaction (the bilayer interferometry experiments), force-dependent dimer-dimer interaction, Ca²⁺ sensitivity, changes wrought by a deafness mutation, and modeling. These observations are all consistent with each other.

The paper presents a provocative suggestion, that the tip link is far more dynamic than we thought otherwise. If the tip-link cadherins behave in vivo as they do in vitro (e.g., no other biological processes modify interaction dynamics), is difficult to argue against the idea that the tip link is remarkably dynamic. That said, the paper would have been strengthened considerably by an in vitro demonstration that their suggestion is correct. The present manuscript is complete and strong, but I guess we will need to wait for future experiments to determine whether the model presented works in an actual hair bundle.

I otherwise have relatively few comments. The paper is beautifully written and illustrated, and the data appear to be extremely reliable, as different experiments nicely reinforce each other (and are consistent with many measurements reported in the literature).

Major comments

1. I have one major point that the authors should address more thoroughly, which is brought up (albeit from a different perspective) in the Discussion, page 16. The authors need to consider a wider range of past data in discussing the implications of the 8 sec tip link lifetime. The question is whether after tip-link dissociation, the PCDH15 dimer ends and the CDH23 dimer ends can possibly be close enough together to drive rebinding. One argument against that idea is that based on experiments like Assad et al. (1991), which show a large positive bundle movement after breaking all the tip links; if the stereocilia pair that formerly housed the tip link in question moves as far as the bundle movements measured, the two ends will likely be far too far apart. However, the large positive movement is a consequence of having all the tip links break at once; if only one breaks, the remaining stereocilia pairs with intact tip links will likely hold the stereocilia pair with a broken tip link in place, given the coherence of the bundle. (The authors could make this point.)

We have expanded our discussion to address the points made by the reviewer in Pages 17-19. We appreciate the point made about the mechanical coherence of the bundle and its potential role in holding stereocilia sufficiently close together to promote reformation. We now include these arguments in the discussion.

However, there is at least one more challenge. Those same experiments and others indicate that the tip link is extended at rest, and the rest extension has been calculated to be on the order of 10-15 nm. Moreover, there are data that suggest that the CDH23 insertion point climbs up somewhat after tip links break (Kazmierczak et al., 2007), which make the separation even greater. At some point the tip link ends waving in the breeze will never find each other as their bases are too far apart.

We now include a short discussion of the elasticity problem on Page 16:

"Although tip links will be extended at rest by at least a few nanometers due to their intrinsic elasticity, making the possibility of the ends snapping away from each other after unbinding likely, random thermal fluctuations in the stereocilia and in the tip links themselves will act to increase the likelihood of rebinding."

Although there are data to suggest that the CDH23 insertion climbs after tip links are broken, we wonder how reliable they are. We note on Page 16 that extracellular Ca²⁺ chelators have detrimental effects beyond tip-link rupture, including disassembly of the actin core within shorter rows of stereocilia and removal of most lateral links¹⁻³. The disassembly of actin would shorten the lower stereocilia, which might appear as climbing of insertion points. Additionally, if tip links are broken and exposed to a low Ca²⁺ solution before fixation and antibody staining, decreased Ca²⁺ occupancy between EC domains will cause the protein to become softer and its structure will tend to form more of a random coil. Indeed, the Kazmierczak et al., 2007 paper demonstrates this effect rather well in their TEM images of tip link proteins in Ca²⁺-free conditions. Since the epitope used to show that CDH23 "climbs" up the stereocilia is at the very N-terminus of CDH23 (antibody PB264, raised against amino acids 123-135), and since staining with this

antibody requires the removal of Ca^{2+} (since it is raised against a Ca^{2+} -binding motif), it is possible that the effect is largely due to the antibody epitope moving closer the edge of the upper stereocilium due to the Ca^{2+} -dependent softening of the CDH23 molecules. Later work from the Kachar group showed via co-staining for several proteins within the upper tip link density (MYO7A, harmonin, sans, and CDH23) that disruption of tip links with Ca^{2+} -chelators caused no disruption or movement of the UTLD⁴. We cite this paper and mention the lack of UTLD movement on Page 14.

Based on what we know about the hair bundle, a tip-link lifetime of 8 sec under resting conditions would suggest that starting with a bundle with intact tip links, the bundle would rapidly lose transduction as tip links break but are not re-engaged. The authors should explain more thoroughly why they expect that tip links will reform after breaking. (It is possible that new CDH23 dimers are recruited, but then why does the upper insertion point appear to be a stable structure? Maybe it isn't.)

We now include a more thorough discussion of why we think tip links should reform in the discussion. We were pleased to see a recent preprint from the Hudspeth group that appears to show evidence of rapid rebinding of tip links in bullfrog and rat hair cells after gentle Ca^{2+} chelation⁵, and we now include this reference in the discussion. These data appear to indicate that at least some tip links have a stable upper insertion point.

2. The argument at the bottom of the page about the concentration of PCDH15 or CDH23 fragments necessary for inhibition of reformation is circular. The authors suggest that one needs to use at least $0.34 \mu\text{M}$ of fragments, as that is the calculated free-end concentration. But the calculation of the free-end concentration assumes that if tip links break rapidly (i.e., 8 sec lifetime), they reform rapidly.

The authors also dismiss the idea that the fragment-blocking experiment can be done because the proteins are unstable at high concentrations. Yet they describe experiments where they use $>1 \mu\text{M}$ tip-link fusion proteins in coupling reactions with DNA tethers for 1.5 hr. It seems like all the ingredients for doing the experiments are in the authors' hands—one would expect then that applying $>1 \mu\text{M}$ tip-link protein to a quiescent hair-cell prep should cause a rapid loss of transduction, with a half life on the order of seconds. It is a shame that the authors did not perform this experiment as it would have nailed their model for the dynamic nature of the tip-link bond.

We now clarify this point in the discussion (Pages 16-17). We note that $0.3 \mu\text{M}$ represents the theoretical 50% equilibrium bound case, whereas given the average number of engaged transduction channels in a hair bundle, the equilibrium bound probability is likely $\sim 95\%$ ⁶, equivalent to an effective concentration of $\sim 10\text{-}15 \mu\text{M}$. Preliminary kinetic modeling suggests that application of $1 \mu\text{M}$ free tip link protein will have almost no effect on the average amount of transduction.

Other comments

Page 2, first sentence of second paragraph. While it is true that the tip link has been shown by EM to be a double-stranded complex, given the far greater molecular information we have now as compared with the year 2000, I recommend modifying this description. More accurately it is an antiparallel structure of one PCDH15 dimer and one CDH23 dimer, and this first sentence should reflect that.

We have now modified this description to more accurately reflect the antiparallel structure of the tip link.

Page 5, line 5. The increase was sublinear, not "slower than linear" (what does that mean, actually?). Also, for this point, what data show this sublinear behavior? Point to them.

We have clarified this point in the text: "The most probable unbinding force increased linearly with the logarithm of the loading rate, indicating that force also accelerated the unbinding rate (Fig. 1f)."

Page 5, line 10 and elsewhere. The SI unit of time is the second, and it is abbreviated "s". Please change " 1.9 ± 0.4 seconds" to " 1.9 ± 0.4 s". The inverse time unit is s^{-1} , which the authors use correctly.

We have now fixed this error. We now specifically use "s" when describing a measurement and "seconds" in our discussions.

Page 5, lines 11-12. Please explain how the f_β measurement was extracted from the data in Fig. 1f.

We now explain in the text that f_β is a fit parameter of the Bell-Evans model, which was used to fit unbinding data. In the methods section (Page 37) we now define f_β as:

$$f_\beta = \frac{k_B T}{x_{ts}}$$

where k_B is the Boltzmann constant, T is absolute temperature, and x_{ts} is the distance from the bound state to the transition state for unbinding.

We also now explain the fit parameters clearly in the legend of Fig. 1f.

Page 7, line 4. Please write " 3.46 ± 0.70 " or " 3.5 ± 0.7 "—it does not make sense that the precision of the mean is greater than the precision of the error.

We have now corrected this and all significant figures in the manuscript.

Page 14, last sentence. It is a little funny to say bath applied or paired with a stimulus. It's like apples and giraffes. The chelators are usually bath applied or applied by iontophoresis, which are significantly different approaches, and they are either applied to quiescent bundles (usually true with bath application) or bundles undergoing mechanical stimuli (usually true with iontophoresis). But don't mix those different conditions—the naïve reader will likely be confused as to the meaning.

We agree that this might be confusing for the naïve reader and have removed this description.

Page 15, second sentence. I am puzzled as to why Zhao et al. is cited for this concern (i.e., that chelation coupled with force could lead to tip link unfolding). Zhao et al. applied chelators to quiescent bundles.

We agree, and we have removed this citation.

Page 16, last paragraph. 300 nM is the same (more or less) as 0.34 μ M, not an order of magnitude lower. The Lelli et al. (2010) paper reports a half-maximal inhibitory concentration of CDH23 fragment of 40 nM, not 300 nM. Thus while the order of magnitude difference is correct, the concentration reported is not. That said, 40 nM is just the half-maximally effective concentration. The Lelli et al. paper is vague as to what conditions were used in any attempts to block transduction in the absence of chelator treatment—their Fig. 5B indicates that they used a concentration as high as 200 nM in the post-chelator experiments, at least.

This was a typo, we meant to write ~ 10 -15 μ M, the concentration "likely to be present for rapid rebinding" in a physiological condition. We agree that the Lelli et al. paper is vague in what conditions they used to block transduction in the absence of chelator treatment, although we interpreted their description to mean the maximal concentration applied in the post-chelator experiments, which is 300 nM.

Page 19, last line. Change to "2 mM (pink) Ca²⁺."

Changed.

Page 21. Move the description of the bilayer interferometry method from 2a to 2c, where the experiment is described.

Moved.

Page 22. First line of 3a legend. What does "These" refer to?

"These" referred to dimer unbinding forces (black circles). We have now altered this description for clarity.

Signed, Peter Barr-Gillespie

We greatly appreciate the thoughtful and detailed comments!

Reviewer #2 (Remarks to the Author):

This manuscript carefully measures the dynamics and strength of Tip-link connection of single-bond, double-stranded, and hereditary deafness mutation cases under various extracellular Ca^{2+} , with single-molecule force spectroscopy (under force condition) as well as ensemble bilayer-interferometer method (under no force condition). With a derived kinetic model considering a force-dependent off-rate of the PCDH15-CDH23 bond and a force-dependent effective concentration C_{eff} , the manuscript calculates a relation, which can't be measure directly, between average life-time of bond and applied force. Furthermore, Monte Carlo simulation was conducted in this manuscript to understand the behavior of tip-link connection in oscillating hair bundles in relation to physiological force range. Overall, this manuscript presents a complete and sound research project with both single-molecule results, ensemble measurements, theory calculation and simulation; these results provide significant foundation for further studies in related field; hence the reviewer suggests to accept this manuscript to publish on Nature Communication with minor revision:

1) Rupture force of PCDH15-CDH23 bond is measured with a configuration showed on Fig. 1c by dual optical traps. How to the authors know the measured rupture force is resulting from the breaking of PCDH15-CDH23 bond, rather than from other bonds or tethers, i.e: poor DNA tethers breaks or unbinds from the beads? Or how do the authors confirm that the data of broken DNA tethers is not mixed into the data of rupture force from PCDH15-CDH23 bond?

Our tethers are anchored covalently to the beads except for a biotin-streptavidin linkage and a 2.4 kb dsDNA tether. The most probable rupture force for the biotin-streptavidin interaction is well above that of any of the measurements we made (Fig. 1), so we are able to eliminate the possibility of this obscuring our results.

Fig. 1 Unbinding force as a function of loading rate for full-length dimers (black circles) and single-bonds (magenta squares) in 2 mM Ca^{2+} , as shown in Fig. 3A. Overlaid is a fit of the most probable rupture force distribution of the biotin-streptavidin interaction from Sedlack et al., 2017⁷, corrected for 4 biotin-streptavidin bonds in series.

The dsDNA tethers were produced by PCR amplification and verified for size using agarose gel electrophoresis. Breaking this component of the tether would require shearing of the entire 2.4 kb tether. Shearing of long DNA tethers within the domain of loading rates that we explored typically happens around or above 65 pN⁸⁻¹⁰, which is above the highest single rupture force we measured. The domain of rupture force measurements was kept within specific loading rate domains for this reason. Additionally, DNA tethers of this length and pulled in this orientation also undergo an overstretching transition which has a distinct mechanical signature. This was not observed in our data.

2) While stretching protein-DNA tethers (Fig. 1c and Fig. 1d), how does PCDH15-CDH23 tip-link behave before rupture, is it similar as a spring? In case the tip-link was stretched for some distance in the experiments, how does it compare with physiological moves of hair bundle due to sound pressure?

The PCDH15-CDH23 tip link does behave as a spring. Bartsch et al., 2019 measured the stiffness of 11 tip-link EC domains of PCDH15 using optical tweezers to be 1-2 pN nm⁻¹ between 0 and 20 pN¹¹. This corresponds to a stiffness of ~10-20 pN nm⁻¹ for a single EC domain. These measurements are surprisingly consistent with our model in the same Ca^{2+} conditions, which calculates EC domain stiffness to be ~12 pN nm⁻¹.

Our experiments were performed over a range of forces—and therefore of tip-link extensions—which represent those within the physiological range. Sensitive experiments in bullfrog saccular hair cells suggest that transduction channels open with a force of ~5 pN above resting tension, resulting in a operating range of ~10-15 pN^{12,13}. Hair bundle movements measured in mammalian excised cochlear preparations appear to be on the order of 10 nanometers in response to a 60 dB noise stimulus¹⁴. We discuss these concepts on Page 12. These forces and deflections are similar to those in vivo and are explored in both our force spectroscopy experiments and in our Monte-Carlo modeling in Fig. 6.

3) In Fig. 1f, could you please explain and discuss more on X-intercept, where the unbinding force is 0pN? How to relate this number with ensemble measurements?

We fit single-bond unbinding data with the Evans-Ritchie model, where the most probable unbinding force f^* is defined as:

$$f^* = f_\beta \cdot \ln \frac{r_l}{f_\beta k_{off}^0}$$

where f^* is the most probable rupture force, f_β is the force scale, r_l is the force loading rate, and k_{off}^0 is the off rate at zero force.

We now note in the methods section (Page 37) that the force scale f_β is defined as:

$$f_\beta = \frac{k_B T}{x_{ts}}$$

where k_B is the Boltzmann constant, T is absolute temperature, and x_{ts} is the distance from the bound state to the transition state for unbinding.

k_{off}^0 and x_{ts} are the characteristic properties for a given molecular pair, and are the two fit parameters in the equation. k_{off}^0 is the same zero-force off-rate that one would measure in an ensemble or bulk biochemical experiment. Since $k_B T$ is constant at a given temperature, we use f_β as the fit parameter. The slope of the line fit in Fig. 1f is determined by f_β , and the x-intercept is determined by the off rate, $k_{off}^0 \cdot f_\beta$. So, for the pink line in Fig. 1f, the slope is $f_\beta = 13.5$ and the x-intercept is $k_{off}^0 \cdot f_\beta = 13.5 \cdot 0.5 = 6.75$.

We now expand upon this point in the Fig. 1f legend.

4) Fig. 2c presents association signal as a function of time, which is measured from bilayer interferometry method. Does Fig. 2c show single-molecule data or averaged ensemble data? Does this signal result from U \diamond B2, or B1 \diamond B2? If the signal is from B1 \diamond B2, how does this data been measured? In case the association signal is from U \diamond B2, there should be a state conversion from U \diamond B2 in Fig. 2b?

Fig. 2c is a single trace from a single sensor in a bilayer interferometry experiment. This experiment is not a single molecule experiment, but is rather the ensemble signal from many hundreds or thousands of molecules interacting with the sensor surface.

The signal in Fig. 2c gives only the *effective* on-rate $k_{on} = 3.5 \pm 0.7 \times 10^5 \text{ M}^{-1} \text{ s}^{-1}$ and the overall off-rate of the connection $k_{off} \approx 0.016 \text{ s}^{-1}$. The reviewer is correct that the association signal as a function of time in Fig. 2c is proportional to the occupancy of the B1 and B2 states. While we cannot directly measure the rates B1 \rightarrow U or B2 \rightarrow B1 from bilayer interferometry, we can calculate these rates using known, intrinsic single-bond kinetics. The B1 \rightarrow U rate in this Ca^{2+} concentration is just the off rate measured in Figure 1f ($k_{off}^0 = 0.5 \pm 0.1 \text{ s}^{-1}$). We also know the single-bond on rate k_{on} , measured by Choudhary and colleagues¹⁵ to be $6.2 \pm 2.8 \times 10^4 \text{ M}^{-1} \text{ s}^{-1}$, and which we might estimate to be one half of the effective dimer on rate measured with BLI. Since we know the overall lifetime of the connection ($\tau = 1/k_{off} = 63.3 \pm 3.8 \text{ s}$), we can calculate C_{eff} from an analytically-derived two-state avidity equation at zero force, derived from our model (from Methods, Pages 37-38):

$$\tau = \frac{C_{eff} k_{on} + 3k_{off}}{2k_{off}^2}$$

The reviewer is correct that in the case of the association signal, there should also be a state conversion from U \rightarrow B1, denoted by $2k_{on}$. We have now clarified in the figure legend that the state diagram refers specifically to the dissociation phase.

5) In Fig. 3a, is the green dashed line ($C_{eff} = \text{constant}$) indicating the local effective concentration at zero force C_{eff0} ? If yes, how to understand that black solid line ($C_{eff} = C_{eff}(F)$) is higher than green dashed line ($C_{eff} = \text{constant}$) in regions of low loading rate ($< 100 \text{ pN s}^{-1}$)? To my understanding, the $C_{eff} = C_{eff}(F)$ should not be higher than that of C_{eff0}

The reviewer's intuition is correct that the constant concentration model should be stronger than the $C_{eff} = C(F)$ model, however the most probable rupture force (or Unbinding Force as labeled in the plot) is not always the best metric for bond strength. The reason that the $C_{eff} = \text{constant}$ line falls below the $C_{eff} = C(F)$ is because of a difference in the shape of the rupture probability density function in each model.

Below is rupture probability density function as a function of time for each model during a 0.2 pN s^{-1} force loading rate. This is the rate where the $C_{\text{eff}} = \text{constant}$ line falls the farthest below the $C_{\text{eff}} = C(F)$ in Fig. 2a. We performed Monte-Carlo simulations for each model to ensure this was not an artifact from numerically solving the rate equations. Initially at the time zero each model is equivalent because the force is zero. Then as force increases rupture probability starts to increase in the $C_{\text{eff}} = C(F)$ model causing a peak around 30 s, then the rupture probability rapidly drops to zero because the probability of the bond lasting that long is zero. At this point the rupture probability for the $C_{\text{eff}} = \text{constant}$ is still greater than zero because it is still probable for a bond to last this long:

This is better shown by examining the fraction bound as a function of time for each model. Here it can be seen that at all times the fraction bound for the $C_{\text{eff}} = \text{constant}$ model is always higher:

6) In Fig.4, the reviewer suggests to plot Fig. 4a' and Fig. 4b' with the bilayer interferometry measurement, similar with Extended Data Fig.3, which is easier for reader to compare both measurements.

We understand and appreciate the reviewer's suggestion. However, when we plotted these data points onto the graph, we found the points too small to be easily viewed by the reader and we worried about these points obscuring the y-intercepts of these plots.

7) On page 36 and 37, the most probable unbinding force of the dimeric interaction is calculated by finding the most probable unbinding time and multiplying it by the loading rate. The unbinding force is a most probable rupture force. In Fig.3a, Fig.4a and Fig.4b, the measured unbinding force data fits very well with calculated most probable rupture force. Could the authors say more about this good fit?

Thank you for noticing the goodness of fit. We were pleased that our simple model captured the essential features of the data over a wide range of experimental conditions (e.g. mutations, Ca^{2+} concentrations) and independent assays (e.g. force spectroscopy, bilayer interferometry). Furthermore, it was encouraging that the values we obtained from the fitting parameters were physically reasonable (e.g. protein stiffness, effective concentration for rebinding), giving us additional confidence in the mechanistic interpretation of the data.

8) In main figures, the x-tick and y-tick are not consistent (i.e.: Fig.4b and Fig.4c). Some are in and some are out. Please adjust ticks.

We have now adjusted ticks to be inward in all figures except for Figs. 1e and 2c. We felt that inward facing ticks in these two figures obscured data sufficiently to warrant outward ticks.

References

1. Zhao, Y., Yamoah, E.N. & Gillespie, P.G. Regeneration of broken tip links and restoration of mechanical transduction in hair cells. *Proc Natl Acad Sci U S A* **93**, 15469-74 (1996).
2. Velez-Ortega, A.C., Freeman, M.J., Indzhykulian, A.A., Grossheim, J.M. & Frolenkov, G.I. Mechanotransduction current is essential for stability of the transducing stereocilia in mammalian auditory hair cells. *Elife* **6**(2017).
3. Indzhykulian, A.A. et al. Molecular remodeling of tip links underlies mechanosensory regeneration in auditory hair cells. *PLoS Biol* **11**, e1001583 (2013).
4. Grati, M. & Kachar, B. Myosin VIIa and sans localization at stereocilia upper tip-link density implicates these Usher syndrome proteins in mechanotransduction. *Proc Natl Acad Sci U S A* **108**, 11476-81 (2011).
5. Alonso, R.G., Tobin, M., Martin, P. & Hudspeth, A.J. Fast recovery of disrupted tip links induced by mechanical displacement of hair bundles. *bioRxiv*, 2020.10.02.324111 (2020).
6. Fettiplace, R. & Kim, K.X. The physiology of mechano-electrical transduction channels in hearing. *Physiol Rev* **94**, 951-86 (2014).
7. Sedlak, S.M. et al. Monodisperse measurement of the biotin-streptavidin interaction strength in a well-defined pulling geometry. *PLoS One* **12**, e0188722 (2017).
8. King, G.A. et al. Revealing the competition between peeled ssDNA, melting bubbles, and S-DNA during DNA overstretching using fluorescence microscopy. *Proc Natl Acad Sci U S A* **110**, 3859-64 (2013).
9. van Mameren, J. et al. Unraveling the structure of DNA during overstretching by using multicolor, single-molecule fluorescence imaging. *Proc Natl Acad Sci U S A* **106**, 18231-6 (2009).
10. Hatch, K., Danilowicz, C., Coljee, V. & Prentiss, M. Demonstration that the shear force required to separate short double-stranded DNA does not increase significantly with sequence length for sequences longer than 25 base pairs. *Phys Rev E Stat Nonlin Soft Matter Phys* **78**, 011920 (2008).
11. Bartsch, T.F. et al. Elasticity of individual protocadherin 15 molecules implicates tip links as the gating springs for hearing. *Proc Natl Acad Sci U S A* **116**, 11048-11056 (2019).
12. Cheung, E.L. & Corey, D.P. Ca²⁺ changes the force sensitivity of the hair-cell transduction channel. *Biophys J* **90**, 124-39 (2006).
13. Jaramillo, F. & Hudspeth, A.J. Displacement-clamp measurement of the forces exerted by gating springs in the hair bundle. *Proc Natl Acad Sci U S A* **90**, 1330-4 (1993).
14. Chen, F. et al. A differentially amplified motion in the ear for near-threshold sound detection. *Nat Neurosci* **14**, 770-4 (2011).
15. Choudhary, D., Kumar, A., Magliery, T.J. & Sotomayor, M. Using thermal scanning assays to test protein-protein interactions of inner-ear cadherins. *PLoS One* **12**, e0189546 (2017).

Reviewer #1 (Remarks to the Author):

I am completely satisfied with the authors' responses to my comments and those of the other reviewer.

Peter Barr-Gillespie

Reviewer #2 (Remarks to the Author):

This manuscript reports measurement of the dynamics and strength of Tip-link connection of single-bond, double-stranded, and hereditary deafness mutation cases with combination of single-molecule results, ensemble measurements, theory calculation and simulation, which together present complete and sound research. After carefully revising and addressing the questions pointed out by referees' comments and/or questions, the manuscript is almost ready to be published. My questions and comments have been (mostly) answered and explained with extra data, figures and references.

Nevertheless, there is still one question/ concern about the single-molecule method used in the manuscript (Fig. 1C). DNA tethers were used to pull Tip-link connections of CDH23-PCDH15 apart with dual optical traps, by which the rupture force of Tip-link were measured with various loading rates. The DNA tethers however appear quite easy to break, potentially due to nicks in the DNA template. The rupture from a DNA tether breaking, rather than from Tip-link unbinding, might result in incorrect data. In single molecule experiments DNA tethers often break before overstretching over force range from 20pN to 60pN, which overlaps exactly with the reported rupture force of single-bond Tip-link (15pN~60pN, Fig.1). In addition, different loading rates might be a factor causing nicked-DNA to break under various forces.

Therefore, the questions is how do the authors double-check the quality of DNA tethers? And the related question, how do the authors confirm that the reported rupture data is resulting from PCDH15-CDH23 unbinding rather than from DNA tethers breaking? And what are the criteria to select the unbinding events from such potentially mixed dataset?

REVIEWER COMMENTS

Reviewer #1 (Remarks to the Author):

I am completely satisfied with the authors' responses to my comments and those of the other reviewer.
Peter Barr-Gillespie

Reviewer #2 (Remarks to the Author):

This manuscript reports measurement of the dynamics and strength of Tip-link connection of single-bond, double-stranded, and hereditary deafness mutation cases with combination of single-molecule results, ensemble measurements, theory calculation and simulation, which together present complete and sound research. After carefully revising and addressing the questions pointed out by referees' comments and/or questions, the manuscript is almost ready to be published. My questions and comments have been (mostly) answered and explained with extra data, figures and references.

Nevertheless, there is still one question/ concern about the single-molecule method used in the manuscript (Fig. 1C). DNA tethers were used to pull Tip-link connections of CDH23-PCDH15 apart with dual optical traps, by which the rupture force of Tip-link were measured with various loading rates. The DNA tethers however appear quite easy to break, potentially due to nicks in the DNA template. The rupture from a DNA tether breaking, rather than from Tip-link unbinding, might result in incorrect data. In single molecule experiments DNA tethers often break before overstretching over force range from 20pN to 60pN, which overlaps exactly with the reported rupture force of single-bond Tip-link (15pN~60pN, Fig.1). In addition, different loading rates might be a factor causing nicked-DNA to break under various forces.

Therefore, the questions is how do the authors double-check the quality of DNA tethers? And the related question, how do the authors confirm that the reported rupture data is resulting from PCDH15-CDH23 unbinding rather than from DNA tethers breaking? And what are the criteria to select the unbinding events from such potentially mixed dataset?

We agree with the reviewer that ensuring rupture of DNA tethers is not a confounding factor in our data is important. The force at which they break does depend on the degree of nicking—more specifically the length of the overlaps at the two 5' ends from which we are exerting force or the distance between nicks on opposite strands. But, relatively short DNA tethers produced by PCR^{1,2} or from oligonucleotide tiling of the m13 phage^{3,4} are common, and there is little evidence in the literature of deleterious effects from nicking using these methods, particularly when they are used with the appropriate level of care (e.g. checking the quality of handles and coupling using gel electrophoresis, minimizing unnecessary hydrodynamic stresses on long strands of DNA during pipetting through proper technique and the appropriate selection of pipette tips, using DNase-free lab supplies, etc., as we now note in the Methods section). We have extensive experience with using DNA tethers for force spectroscopy as we have used them in previous projects²⁻⁴, and we have not had issues with mechanical fragility due to the presence of nicks. Furthermore, we note that the DNA tethers that we used in this study were much shorter than the long lambda DNA tethers often used in single-molecule force measurements (~2.4 kbp vs. 48.5 kbp, or about ~20x shorter), making our tethers far less susceptible to mechanical damage during handling, and less likely to acquire multiple nicks.

For nicking to have a detrimental effect on the stability of the DNA tethers below 60-65 pN, the nick needs to be within ~60 bp of a 5' end⁵, or two nicks need to form on opposite strands within about ~60 bp of each other. The mechanical properties of DNA oligonucleotide shearing are well described, and perhaps best represented by data from Strunz, et al. in 1999. Here, the authors measured the shearing forces of DNA oligonucleotides of various lengths, and describe how the transition state distance x_{ts} and the thermal off rate v scale with the number of base pairs⁶. The off rate at zero force v as a function of the number of base pairs n is:

$$\text{Eq. 1} \quad v \approx 10^{\alpha - \beta n} \cdot s^{-1}$$

where $\alpha = 3 \pm 1$ and $\beta = 0.5 \pm 0.1$ from a linear regression of shearing data. Similarly, the transition state distance x_{ts} was determined by a linear regression to shearing data:

$$\text{Eq. 2} \quad x_{ts} \approx 7 \text{ \AA} + 0.7n \text{ \AA}$$

where n is the number of base pairs. Using these equations, we can calculate the shearing off-rate k_{off}^0 and the force scale f_b for oligonucleotides of different lengths (Table R1).

DNA length (bp)	x _{ts} (nm)	f _b (pN)	Koff0 (s ⁻¹)
10	1.4	2.9	1E-02
20	2.1	2.0	1E-07
40	3.5	1.2	1E-17
60	4.9	0.8	1E-27

Table R1

Biophysical properties of DNA shearing for oligonucleotides of different lengths, calculated with Eq. 1-2.

For DNA oligonucleotides 10-60 bp in length, the force scale f_b (which determines the slope of the Bell-Evans fit) is between 0.8-2.9 pN (Table R1), around 5-10 times lower than that of the monomer rupture data in Fig. 1f. Consequently, if our rupture forces were dominated by DNA shearing of highly nicked tethers, we would expect to see a slope much shallower than observed, independent of the location of the nicks on the 5' ends. When overlaid onto monomer and dimer rupture data, the difference in force scale between tip-link unbinding and oligonucleotide shearing is apparent (Figure R1).

Figure R1

Bell-Evans plots of DNA shearing of oligonucleotides of lengths between 10-60 bp from parameters in Table R1 (dashed lines) plotted on Figure 3a.

Furthermore, if the tether nicking locations were random—as would be expected if nicks were caused by mechanical action or degradation—one would expect a wide range of different shearing lengths to be represented in the unbinding data. Since collections of each shearing length should each give have a different force scale f_b (and therefore a different slope on the loading rate vs. unbinding force plots) we would expect to have a wide and noisy rupture force distribution that could not easily be fit with a standard Bell-Evans model. Yet our data—both the DFS curves and the force-rupture distributions—are consistent with Bell-Evans' based models.

The specificity of our optical-tweezer force spectroscopy measurements are also supported by our experiments where the tip-link interaction is specifically modified, such as alterations of $[Ca^{2+}]$, point mutations in the binding domains, and the monomer vs. dimer interaction. For example, if our monomer data in Fig. 1c is dominated by the shearing of highly-nicked DNA tethers, we would not expect to see a stark increase in the most-likely rupture force for the dimeric interaction at the same force loading rate, as shown in Fig. 3a and Fig. 4a. The specificity of our force spectroscopy experiments to the tip-link interaction is further supported by independent measurements of kinetics at zero force using biolayer interferometry; if our data were dominated by shearing of nicked tethers, then the derived kinetics would not have been consistent with these independent solution kinetic measurements.

Finally, we performed a subset of our dimer experiments *without DNA tethers*. As noted in the methods section:

For full-length tip link fusion proteins under some conditions (10 μM Ca^{2+} , R113G mutations, fast loading rates), it was necessary to perform experiments without the DNA tether in order to increase the rate of data collection. For these experiments, fusion proteins were directly biotinylated with BG-Biotin (NEB), buffer exchanged with a Zeba Desalting Column (ThermoFisher) three times, and directly coated onto streptavidin coated beads. Individual tethers were verified using a WLC fit to the force-extension traces⁷ and tether frequency was kept below 1:10 tethers:attempts.

For example, the 95.62 pN s^{-1} loading rate for the full-length tip-link dimer in Fig. 3A was acquired with these direct-biotinylated constructs. If we overlay the most probable unbinding force from a small number of unbinding events taken at this loading rate with DNA tethers not included in the paper ($n = 25$ unbinding events), we see no difference in the most probable unbinding force (Figure R3). This again indicates that our DNA tethers are intact and sufficiently mechanically stable.

Figure R3

Monomer and full-length dimer unbinding data, modified from Fig. 3a. The most probable unbinding force at 95.62 pN s^{-1} for full-length dimer constructs with DNA tethers attached (blue diamond, $n=25$) is overlaid.

References

1. Lipfert, J., Kerssemakers, J.W., Jager, T. & Dekker, N.H. Magnetic torque tweezers: measuring torsional stiffness in DNA and RecA-DNA filaments. *Nat Methods* **7**, 977-80 (2010).
2. Zhang, X., Halvorsen, K., Zhang, C.Z., Wong, W.P. & Springer, T.A. Mechanoenzymatic cleavage of the ultralarge vascular protein von Willebrand factor. *Science* **324**, 1330-4 (2009).
3. Halvorsen, K., Schaak, D. & Wong, W.P. Nanoengineering a single-molecule mechanical switch using DNA self-assembly. *Nanotechnology* **22**, 494005 (2011).
4. Yang, D., Ward, A., Halvorsen, K. & Wong, W.P. Multiplexed single-molecule force spectroscopy using a centrifuge. *Nat Commun* **7**, 11026 (2016).
5. Hatch, K., Danilowicz, C., Coljee, V. & Prentiss, M. Demonstration that the shear force required to separate short double-stranded DNA does not increase significantly with sequence length for sequences longer than 25 base pairs. *Phys Rev E Stat Nonlin Soft Matter Phys* **78**, 011920 (2008).
6. Strunz, T., Oroszlan, K., Schafer, R. & Guntherodt, H.J. Dynamic force spectroscopy of single DNA molecules. *Proc Natl Acad Sci U S A* **96**, 11277-82 (1999).
7. Bouchiat, C. et al. Estimating the persistence length of a worm-like chain molecule from force-extension measurements. *Biophys J* **76**, 409-13 (1999).

Reviewer #2 (Remarks to the Author):

I think all the remaining questions have been address well. I think this intersting manuscript is ready for publication.